# Loss of ZNRF3/RNF43 unleashes EGFR in cancer

Fei Yue[1,2]*, Amy T Ku[1], Payton D Stevens[3,4], Megan N Michalski[3], Weiyu Jiang[1], Jianghua Tu[5], Zhongcheng Shi[6], Yongchao Dou[1], Yi Wang[7], Xin-Hua Feng[8], Galen Hostetter[9], Xiangwei Wu[10], Shixia Huang[6,11,12,13], Noah F Shroyer[2,13], Bing Zhang[1,13,14], Bart O Williams[3,9], Qingyun Liu[5], Xia Lin[15], Yi Li[1,11,13,16]*

[1]Lester and Sue Smith Breast Center, Baylor College of Medicine, Houston, United States; [2]Department of Medicine, Baylor College of Medicine, Houston, United States; [3]Department of Cell Biology, Van Andel Institute, Grand Rapids, United States; [4]Biological Sciences Department, Miami University, Oxford, United States; [5]Texas Therapeutics Institute and Brown Foundation Institute of Molecular Medicine, University of Texas Health Science Center at Houston, Houston, United States; [6]Advanced Technology Cores, Baylor College of Medicine, Houston, United States; [7]State Key Laboratory of Proteomics, Beijing Proteome Research Center, National Center for Protein Sciences (Beijing), Beijing Institute of Lifeomics, Beijing, China; [8]Life Sciences Institute, Zhejiang University, Hangzhou, China; [9]Van Andel Institute, Core Technologies and Services, Grand Rapids, United States; [10]Department of Clinical Cancer Prevention, The University of Texas MD Anderson Cancer Center, Houston, United States; [11]Department of Molecular and Cellular Biology, Baylor College of Medicine, Houston, United States; [12]Department of Education, Innovation and Technology, Baylor College of Medicine, Houston, United States; [13]Dan L. Duncan Comprehensive Cancer Center, Baylor College of Medicine, Houston, United States; [14]Department of Molecular and Human Genetics, Baylor College of Medicine, Houston, United States; [15]The First Affiliated Hospital of Zhejiang University, Hangzhou, China; [16]Department of Molecular Virology and Microbiology, Baylor College of Medicine, Houston, United States

*For correspondence:
fy2111@nyu.edu (FY);
liyi@bcm.edu (YL)

Competing interest: The authors declare that no competing interests exist.

## eLife Assessment

This manuscript presents **solid** evidence suggesting that the loss of ZNRF3 and RNF43, two E3 ubiquitin ligases, leads to dysregulation of EGFR signaling in cancer. The authors propose that EGFR is a direct substrate of ZNRF3/RNF43. While the authors provide immunoprecipitation data showing increased detection of ubiquitinated species, this evidence does not definitively establish that EGFR itself is ubiquitinated by RNF43/ZNRF3. The absence of direct evidence for EGFR ubiquitination is a major limitation, although the findings are **useful** as they may provide novel insights into the mechanisms underlying EGFR-driven cancers and open new therapeutic avenues.

**Abstract** ZNRF3 and RNF43 are closely related transmembrane E3 ubiquitin ligases with significant roles in development and cancer. Conventionally, their biological functions have been associated with regulating WNT signaling receptor ubiquitination and degradation. However, our proteogenomic studies have revealed EGFR as the protein most negatively correlated with *ZNRF3/RNF43* mRNA levels in multiple human cancers. Through biochemical investigations, we demonstrate that ZNRF3/RNF43 interact with EGFR via their extracellular domains, leading to EGFR

ubiquitination and subsequent degradation facilitated by the E3 ligase RING domain. Overexpression of *ZNRF3* reduces EGFR levels and suppresses cancer cell growth in vitro and in vivo, whereas knockout of *ZNRF3/RNF43* stimulates cell growth and tumorigenesis through upregulated EGFR signaling. Together, these data suggest ZNRF3 and RNF43 as novel E3 ubiquitin ligases of EGFR and establish the inactivation of ZNRF3/RNF43 as a driver of increased EGFR signaling, ultimately promoting cancer progression. This discovery establishes a connection between two fundamental signaling pathways, EGFR and WNT, at the level of cytoplasmic membrane receptors, uncovering a novel mechanism underlying the frequent co-activation of EGFR and WNT signaling in development and cancer.

## Introduction

Zinc And Ring Finger 3 (ZNRF3) and Ring Finger Protein 43 (RNF43) are two closely related single-pass transmembrane E3 ligases with significant roles in embryonic development, tissue homeostasis and regeneration, and diseases (*Hao et al., 2012*; *Koo et al., 2012*; *Planas-Paz et al., 2016*; *Szenker-Ravi et al., 2018*; *Basham et al., 2019*; *Lee et al., 2020*; *Sun et al., 2021*). Extensive research has demonstrated their involvement in critical developmental processes such as limb development (*Szenker-Ravi et al., 2018*), liver zonation (*Planas-Paz et al., 2016*), and mammalian sex determination (*Harris et al., 2018*). These two enzymes also function as tumor suppressors, as evidenced by the promotion of intestinal and adrenal hyperplasia (*Koo et al., 2012*; *Basham et al., 2019*) and hepatocellular carcinogenesis (*Mastrogiovanni et al., 2020*) in mice upon tissue-specific inactivation of *Znrf3/Rnf43*. Similarly, *Rnf43* deficiency leads to thickened mucosa, hyperplasia, and cellular atypia in the stomach (*Neumeyer et al., 2019*), and enhanced tumor growth in a mouse model of inflammatory colorectal cancer (*Eto et al., 2018*). Conversely, overexpression of ZNRF3 and RNF43 suppresses cancer cell proliferation, migration and invasion, and drug resistance in multiple human cancer cell lines (*Jiang et al., 2013*; *Zhou et al., 2013*; *Qiu et al., 2016*; *Pangestu et al., 2021*; *Radaszkiewicz et al., 2021*). Importantly, in cancer patients, ZNRF3 and RNF43 are frequently inactivated by gene deletion or loss-of-function mutations (*Hao et al., 2016*; *Yu et al., 2020*). For instance, *ZNRF3* exhibits the highest rate of copy number variations in adrenocortical carcinomas (ACCs), homozygously deleted in approximately 20% of ACCs (*Assié et al., 2014*). *RNF43* is mutated in approximately 15% of endometrial cancer, 12% of stomach cancer, 11% of colorectal cancer, and 7% of pancreatic cancer (*Tu et al., 2019*). These findings highlight the clinical significance of ZNRF3/RNF43 and the importance of fully understanding the molecular mechanism(s) by which they affect development and cancer.

The primary known molecular function of ZNRF3/RNF43 is the regulation of WNT signaling. As transmembrane E3 ligases, they ubiquitinate the WNT signaling receptor Frizzled, targeting it for degradation, thus dampening WNT signaling (*Hao et al., 2012*; *Koo et al., 2012*). Conversely, R-spondins (RSPO1-4) act as antagonistic peptide ligands, binding to ZNRF3/RNF43 and promoting their auto-ubiquitination and membrane clearance, thus maintaining WNT receptor levels and potentiating WNT signaling (*Carmon et al., 2011*; *de Lau et al., 2011*; *Glinka et al., 2011*; *Hao et al., 2012*; *Koo et al., 2012*; *Lebensohn and Rohatgi, 2018*; *Park et al., 2018*; *Szenker-Ravi et al., 2018*; *Dubey et al., 2020*; *Park et al., 2020*). Furthermore, activation of WNT signaling leads to transactivation of *ZNRF3* and *RNF43* expression via β-catenin (*Hao et al., 2012*; *Koo et al., 2012*), creating a negative feedback loop that tightly regulates WNT signaling during normal development and tissue homeostasis. However, in cancer cells, alterations such as recurrent *RSPO* gene fusion or *ZNRF3/RNF43* deletion or mutations disrupt this feedback loop, resulting in hyperactive WNT signaling and subsequent cancer development and progression (*Seshagiri et al., 2012*; *Hao et al., 2016*).

Although most molecular functions of RSPO-ZNRF3/RNF43 have been linked to their modulation of WNT signaling, emerging evidence suggests that ZNRF3 and RNF43 may possess WNT-independent functions. For instance, RSPO and WNT ligands exhibit non-equivalent roles in various processes, including mammary epithelial cell growth (*Klauzinska et al., 2012*), mammary side branches (*Geng et al., 2020*), intestinal stem cell self-renewal (*Yan et al., 2017*), and cochlea development (*Mulvaney et al., 2013*). Furthermore, loss of RNF43 function promotes mouse gastric epithelium proliferation and human gastric cancer cell xenograft growth, without detectable impact on WNT signaling activity (*Neumeyer et al., 2019*; *Neumeyer et al., 2021*). Additionally, the inhibitory effects of RSPO-ZNRF3 on BMP signaling are not abolished by *β-catenin* knockdown (*Lee et al., 2020*). Moreover,

approximately 40% of *ZNRF3* or *RNF43*-mutant colon tumors also harbor alterations in either *APC* or *CTNNB1*, suggesting the involvement of these E3 ligases beyond the regulation of WNT signaling (*Cerami et al., 2012*; *Gao et al., 2013*). This is further supported by a report of a hotspot *RNF43* G659 frameshift mutation that does not appear to affect WNT signaling (*Fang et al., 2022*). However, the extent to which ZNRF3 and RNF43 act on other substrates, apart from WNT receptors, to influence development and cancer has been unclear.

Here, we report a new function of ZNRF3 and RNF43. Our work began with computational studies of public data identifying a strong negative correlation between *ZNRF3/RNF43* mRNA levels and EGFR protein expression in datasets of human ACCs and colorectal cancers. Subsequently, we investigated the impact of depleting *ZNRF3* or *RNF43* on EGFR clearance at the cell surface in mouse embryonic fibroblasts (MEFs) and human cancer cell lines, including *APC*-mutated colorectal cancer cells. Notably, we discovered that ZNRF3/RNF43 interacts with EGFR through their extracellular domains (ECDs), leading to EGFR ubiquitination and degradation via the E3 ligase RING domain. Additionally, we substantiated our findings in multiple cell lines, human cancer cell xenograft models, and genetically engineered mouse models, wherein the loss of *ZNRF3/RNF43* resulted in elevated EGFR levels and facilitated cancer progression.

## Results

### EGFR protein levels negatively correlate with *ZNRF3/RNF43* mRNA expression in multiple human cancers

To elucidate novel signaling pathways regulated by ZNRF3/RNF43 in cancer, we conducted integrative proteogenomic analyses of human cancer datasets using LinkedOmics (*Vasaikar et al., 2018*). Given the critical role of *ZNRF3* in adrenal hemostasis (*Basham et al., 2019*) and its frequent deep deletion in approximately 20% of ACCs (*Assié et al., 2014*), our initial focus was to examine the ACC dataset from TCGA, comprising 92 samples, to identify proteins exhibiting correlations with *ZNRF3* mRNA levels. Among the proteins evaluated by Reverse Phase Protein Array (RPPA), our analysis revealed EGFR protein levels to be most negatively correlated with *ZNRF3* mRNA levels (r=–0.50 and p=4.5e-4) (*Figure 1A*, *Figure 1—figure supplement 1A*). Notably, ACC tumors with deep deletion in *ZNRF3* exhibited the highest EGFR protein levels compared to ACC tumors with other alterations in the *ZNRF3* locus (*Figure 1B*). These findings suggest that disruption of the gene function of *ZNRF3* may lead to upregulation of EGFR in ACCs.

Besides frequent deletions in ACC, *ZNRF3,* and *RNF43* are also important tumor suppressors in the more commonly occurring colorectal cancer (CRC) (*Koo et al., 2012*; *Bond et al., 2016*); thus, we next analyzed the TCGA colorectal adenocarcinoma (n=629) and CPTAC colon adenocarcinoma (n=110) datasets. By incorporating RPPA-based (TCGA CO/READ) and mass spectrometry-based (CPTAC COAD) proteomic data, we also confirmed EGFR protein to be the one most negatively correlated with *ZNRF3* and *RNF43* mRNA expression in CRC (*Figure 1C*, *Figure 1—figure supplement 1B–E*). Of note, we utilized *ZNRF3/RNF43* mRNA abundances rather than protein levels due to the lack of protein measurements for these two low-abundance enzymes in the current proteomic datasets. Mutations in *ZNRF3/RNF43* are found in approximately 15% of CRCs overall, with high frequencies observed in the microsatellite instability-high (MSI-H) subtype (60–80%) (*Giannakis et al., 2014*; *Bond et al., 2016*; *Tu et al., 2019*; *Vasaikar et al., 2019*). In particular, the *RNF43* G659Vfs*41 frameshift mutation accounts for 40–50% of all *RNF43* mutations in multiple MSI-H cancer types (*Tu et al., 2019*). Interestingly, CRCs harboring the *RNF43* G659Vfs*41 mutation exhibited significantly higher levels of EGFR protein compared to *RNF43* wild-type (WT) tumors (*Figure 1D*, *Figure 1—figure supplement 1F*). A similar difference was observed in stomach cancer, which also displayed a high frequency of the *RNF43* G659Vfs*41 mutation (*Figure 1—figure supplement 1G*). Additionally, in the microsatellite stable subtype of CRCs, predominantly expressing WT RNF43, we found a negative association between *RNF43* mRNA and EGFR protein levels (r=–0.36 and p=0.0018) (*Figure 1E*). Using cBioPortal (*Cerami et al., 2012*; *Gao et al., 2013*), we also detected a negative correlation between *ZNRF3/RNF43* expression and EGFR protein in multiple other cancer types, including prostate cancer where *ZNRF3* or *RNF43* is deleted or mutated with a rate of 5% (*Figure 1F*). Collectively, these results suggest that disruption of *ZNRF3* and *RNF43* may upregulate EGFR protein levels in human cancers.

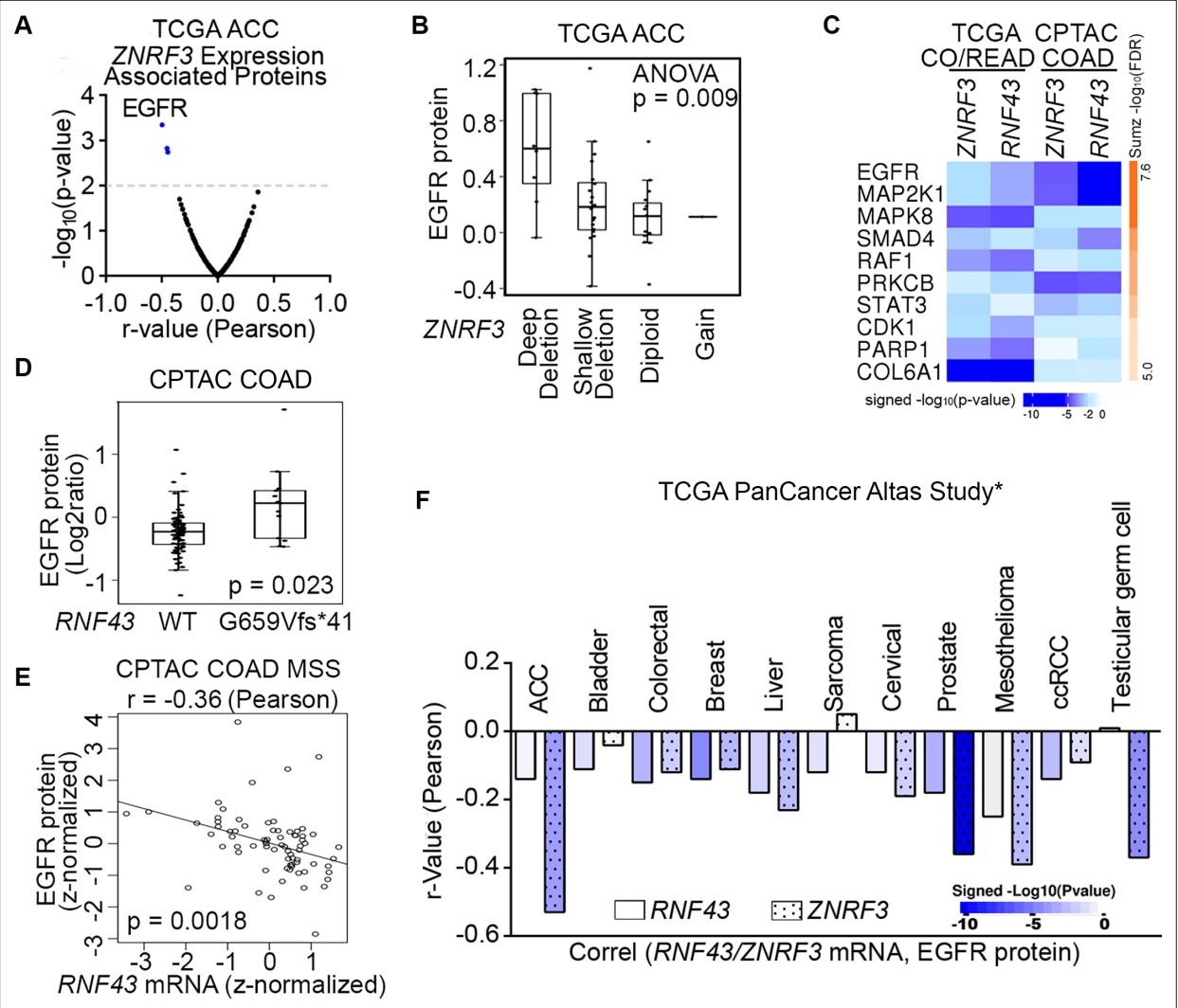

**Figure 1.** Proteogenomic analysis identifies EGFR as the top candidate protein downregulated by ZNRF3/RNF43 in cancers. (**A**) Volcano plot of proteins associated with *ZNRF3* mRNA expression in human adrenal cortical carcinoma (ACC), using the TCGA dataset (n=92). (**B**) Boxplot of EGFR protein levels in human adrenal cortical carcinomas with different *ZNRF3* gene copy number alteration, using the TCGA dataset (n=92). (**C**) The top 10 proteins negatively correlated with *ZNRF3/RNF43* mRNA levels, ranked by Stouffer's method, using the TCGA colorectal adenocarcinoma (CO/READ) (n=629) and the CPTAC colon adenocarcinoma (COAD) (n=110) datasets. (**D**) Boxplot of EGFR protein levels in human colon adenocarcinomas expressing RNF43 WT or G659Vfs*41 mutant, using the CPTAC dataset. (**E**) Scatterplot of EGFR protein level versus *RNF43* mRNA expression using microsatellite stable (MSS) colon adenocarcinoma in the CPTAC dataset. (**F**) Bar graph of significant associations between *RNF43/ZNRF3* mRNA expression and EGFR protein level in cancer datasets from TCGA PanCancer Atlas Study. *, insignificant associations were not shown.

The online version of this article includes the following figure supplement(s) for figure 1:

**Figure supplement 1.** EGFR protein level is negatively associated with *ZNRF3/RNF43* mRNA expression in cancers.

## ZNRF3 and RNF43 downregulate the EGFR protein level

To test whether ZNRF3 and RNF43 regulate EGFR protein, we performed gain-of-function and loss-of-function experiments in several cell lines. In MDA-MB-231, a breast cancer cell line with moderate levels of ZNRF3/RNF43, overexpression of either ZNRF3 or RNF43 substantially reduced EGFR protein levels (*Figure 2A*). Remarkably, ZNRF3 and RNF43 decreased EGFR as robustly as CBL, the best-known E3 ligase of EGFR (*Levkowitz et al., 1998*; *Waterman et al., 1999*). Similarly, in HEK293T cells, ZNRF3/RNF43 overexpression reduced the levels of EGFR (*Figure 2—figure supplement 1A*). As predicted, the phosphorylated form of EGFR declined as well (*Figure 2—figure supplement 1A*). Notably, the reduction in both total EGFR and P-EGFR levels reached an extent comparable to that achieved by CBL (*Figure 2—figure supplement 1A*). Consistent with their known function, ZNRF3/

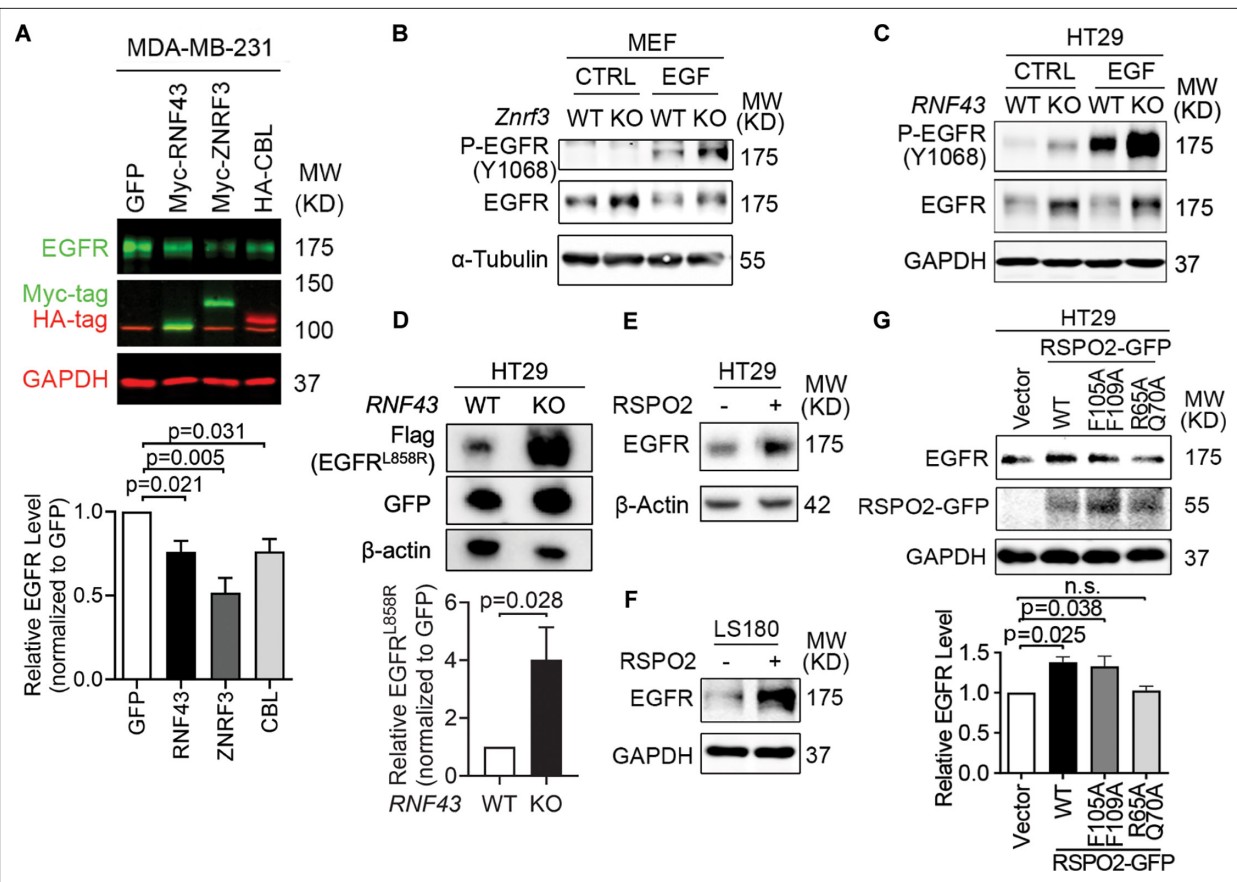

**Figure 2.** ZNRF3/RNF43 downregulates EGFR protein level. (**A**) Overexpression of RNF43, ZNRF3, or CBL decreases EGFR protein level compared to GFP control in MDA-MB-231 cells, shown by representative western blot images (left panel) and quantification results (right panel). Cells were infected with lentivirus expressing GFP or E3 ligases. Means ± SEMs are shown. p-Values were calculated by one-way ANOVA uncorrected Fisher's LSD test. (**B**) *Znrf3* knockout increases P-EGFR and total EGFR levels in murine embryonic fibroblasts (MEFs) upon EGF stimulation (50 ng/ml, 10 min). (**C**) *RNF43* knockout increases P-EGFR and total EGFR levels in HT29 cells untreated or treated with recombinant EGF (50 ng/ml, 10 min). (**D**) *RNF43* knockout increases the level of mutated EGFR[L858R] protein in HT29 cells, as shown by representative western blot images (left panel) and quantification results (right panel). Means ± SEMs are shown. p-Values were calculated by Student's t-test. (**E, F**) RSPO2 treatment (50 ng/ml, 2–4 hr) enhances EGFR protein levels in HT29 (**E**) and LS180 (**F**) cells. (**G**) Overexpression of RSPO2 wild-type (WT) or F105A/F109A mutant but R65A/Q70A mutant enhances EGFR protein level, shown by representative western blot images (left panel) and quantification results (right panel). Means ± SEMs are shown. p-Values were calculated by one-way ANOVA uncorrected Fisher's LSD test. n.s., not significant.

The online version of this article includes the following source data and figure supplement(s) for figure 2:

**Source data 1.** Excel file providing the numerical source data to *Figure 2*.

**Source data 2.** PDF files containing the original, labeled blots and gels to *Figure 2*.

**Source data 3.** TIF files of the raw blots and gels to *Figure 2*.

**Figure supplement 1.** ZNRF3/RNF43 negatively regulates EGFR protein level.

**Figure supplement 1—source data 1.** Excel file providing the numerical source data to *Figure 2—figure supplement 1*.

**Figure supplement 1—source data 2.** PDF files containing the original, labeled blots and gels to *Figure 2—figure supplement 1*.

**Figure supplement 1—source data 3.** TIF files of the raw blots and gels to *Figure 2—figure supplement 1*.

RNF43 overexpression reduced the level of their substrate FZD5 (*Hao et al., 2012*; *Koo et al., 2012*; *Figure 2—figure supplement 1A*). Additionally, these two E3 ligases exhibited selectivity in their regulation of growth factor receptors since overexpression of ZNRF3 or RNF43 did not decrease levels of TGFβ receptor I (*Figure 2—figure supplement 1B*) or FGFR1 (*Figure 2—figure supplement 1C*).

To test the impact of *ZNRF3/RNF43* loss on EGFR levels, we compared WT and *Znrf3* knockout (KO) MEFs because the WT MEFs have significant *Znrf3* expression but very minimal *Rnf43* expression (*Lienert et al., 2011*). We found that *Znrf3* KO enhanced both EGFR and P-EGFR levels, both in the

absence and presence of recombinant EGF (*Figure 2B*). To test the effect of *RNF43* KO, we utilized HT29, a CRC cell line that expresses WT *RNF43* at a high level but minimal *ZNRF3*. CRISPR-KO of *RNF43* enhanced EGFR and P-EGFR levels regardless of EGF stimulation (*Figure 2C*). We validated these results using three additional independent *RNF43* sgRNAs (*Figure 2—figure supplement 1D*). Furthermore, since EGFR is mutationally activated in some human cancers, we transfected a common mutant – EGFR L858R – into *RNF43* WT versus KO HT29 cells and compared the resulting protein levels. As shown in *Figure 2D*, EGFR L858R was significantly higher in *RNF43* KO cells than in WT cells, indicating that RNF43 can degrade both WT and mutated EGFR and its loss can enhance signaling of both WT EGFR and its oncogenic mutant. In addition, we extended our analysis to HCC1954, a breast cancer cell line that expresses the EGFR family member HER2. We found that KO of either *ZNRF3* or *RNF43* enhanced the protein levels of HER2, as well as EGFR (*Figure 2—figure supplement 1E*), suggesting that ZNRF3 and RNF43 downregulate the levels of EGFR and its family members and that the loss of these E3 ligases stabilizes this family of receptor tyrosine kinases.

RSPO1-4 are antagonistic ligands of ZNRF3 and RNF43 (*Hao et al., 2012*; *Koo et al., 2012*). They bind to both ZNRF3/RNF43 and LGR4/5/6, resulting in ZNRF3/RNF43 auto-ubiquitination and degradation (*Hao et al., 2012*; *Koo et al., 2012*). Approximately 18% of CRCs harbor amplification or mutations of *RSPO1-4* or *LGR4/5/6*, and 10% of CRCs show recurrent *RSPO2/3* gene fusions (*Seshagiri et al., 2012*; *Seeber et al., 2019*). These genetic alterations potentially lead to the inhibition of ZNRF3/RNF43, thereby activating WNT and EGFR signaling. Therefore, we tested whether RSPO affected EGFR protein levels in CRC. Short-term treatment with recombinant RSPO2 increased EGFR

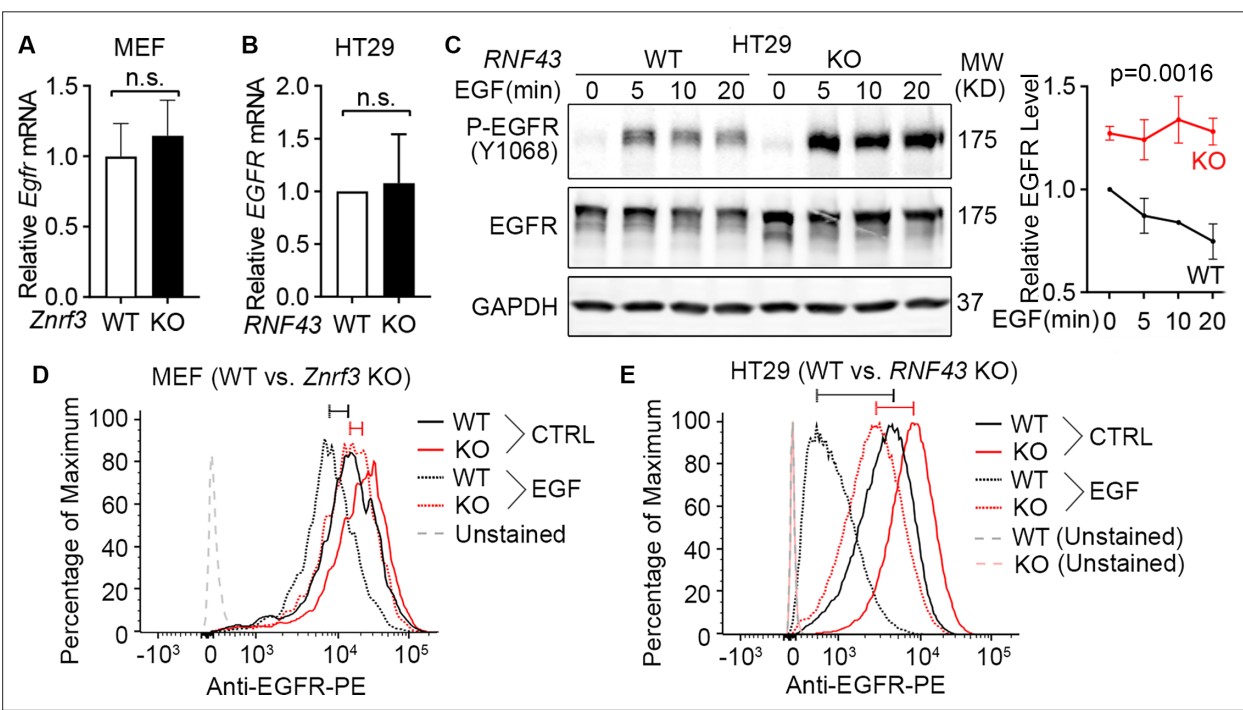

**Figure 3.** Loss of ZNRF3/RNF43 delays EGFR protein degradation. (**A**) Knockout of *Znrf3* has no impact on *Egfr* mRNA level in murine embryonic fibroblasts (MEFs). Means ± SEMs are shown. p-Values were calculated by Welch's t-test. n.s., not significant. (**B**) Knockout of *RNF43* has no impact on *EGFR* mRNA level in HT29 cells. Means ± SEMs are shown. p-Values were calculated by Welch's t-test. (**C**) *RNF43* knockout inhibits EGF-induced EGFR protein degradation in HT29 cells. Cells were stimulated with EGF (50 ng/ml) for indicated times. Representative western blot images (left panel) and quantification results (right panel) were shown. Means ± SEMs are shown. p-Values were calculated by two-way ANOVA uncorrected Fisher's LSD test. (**D**) *Znrf3* knockout increases the cell surface level of EGFR protein in MEFs unstimulated or stimulated with EGF (50 ng/ml, 10 min). (**E**) *RNF43* knockout increases the cell surface level of EGFR protein in HT29 cells unstimulated or stimulated with EGF (50 ng/ml, 10 min). The cell surface EGFR levels were measured by flow cytometry. The bars mark the relative peak shifts after EGF stimulation in WT (wild-type) (black) or KO (knockout) (red) cells.

The online version of this article includes the following source data for figure 3:

**Source data 1.** Excel file providing the numerical source data to *Figure 3*.

**Source data 2.** PDF files containing the original, labeled blots and gels to *Figure 3C*.

**Source data 3.** TIF files of the raw blots and gels to *Figure 3C*.

levels in two CRC cell lines (*Figure 2E and F*) and 293T cells (*Figure 2—figure supplement 1F*). Further, this impact of RSPO2 depended on the presence of intact ZNRF3/RNF43, as RSPO2 failed to elevate EGFR levels further in HT29 cells knocked out for *RNF43*, the predominant one compared to *ZNRF3* (*Figure 2—figure supplement 1G*). Conversely, the RSPO2 R65A/Q70A mutant, which cannot bind to ZNRF3/RNF43 (*Xie et al., 2013*), failed to elevate EGFR levels (*Figure 2G*). On the other hand, the RSPO2 F105A/F109A mutant, which cannot bind to LGRs (*Xie et al., 2013*), still enhanced EGFR levels similar to RSPO2 WT (*Figure 2F*). These data collectively suggest that RPSO2 regulation of EGFR does not rely on LGR-binding but requires its interaction with ZNRF3/RNF43.

## ZNRF3 and RNF43 induce EGFR ubiquitination and degradation

After demonstrating that ZNRF3 and RNF43 regulate EGFR protein levels, we next sought to elucidate the underlying mechanism. We first examined whether ZNRF3 and RNF43 regulate *EGFR* transcripts and found that ZNRF3/RNF43 did not impact *EGFR* mRNA levels in either MEFs or HT29, as determined by quantitative PCR (qPCR) (*Figure 3A and B*). Then, we studied the protein stability of EGFR in WT and *RNF43* KO HT29 cells. We treated cells with recombinant EGF to induce EGFR internalization/degradation and then collected cell lysates at different time points to measure EGFR protein levels by western blotting. We found that KO of *RNF43* delayed EGFR degradation and sustained P-EGFR levels (*Figure 3C*). Since EGFR clearance is initiated at the cell membrane and accelerated by EGF treatment, we also evaluated the impact of ZNRF3/RNF43 on EGFR levels at the cell surface. Flow cytometry revealed that *ZNRF3* KO in MEFs (*Figure 3D*) or *RNF43* KO in HT29 (*Figure 3E*) increased cell surface EGFR in both unstimulated and EGF-stimulated conditions. These data together suggest that ZNRF3 and RNF43 induce cell surface EGFR internalization and degradation, and that their inactivation leads to EGFR accumulation at the cell surface and thus enhanced EGFR signaling.

EGFR degradation is primarily mediated by ubiquitination (*Galcheva-Gargova et al., 1995*; *Levkowitz et al., 1998*). To examine the impact of ZNRF3 and RNF43 on EGFR ubiquitination, we performed EGFR immunoprecipitation (IP) followed by ubiquitin (Ub) immunoblotting (IB). Remarkably, RNF43 overexpression in MDA-MB-231 cells enhanced EGFR ubiquitination as potently as CBL overexpression (*Figure 4A*). Conversely, in *RNF43* KO HT29 cells and MDA-MB-231 cells, anti-Ub IP brought down substantially less EGFR protein (*Figure 4B*, *Figure 4—figure supplement 1A*), and anti-EGFR IP produced much less ubiquitinated forms of EGFR (*Figure 4C*), indicating that *RNF43* KO diminished EGFR ubiquitination. Next, we asked whether the E3 ligase activity of ZNRF3/RNF43 is needed to regulate EGFR ubiquitination. The RING domain is required for ZNRF3/RNF43 E3 Ub ligase function (*Pickart and Eddins, 2004*). Therefore, we compared the impact of ZNRF3/RNF43 versus their RING domain deletion mutants (ΔRING) on EGFR protein levels. We found that both ΔRING mutants failed to decrease EGFR levels (*Figure 4D and E*) or HER2 levels (*Figure 4—figure supplement 1B*). Furthermore, while WT ZNRF3 overexpression increased EGFR ubiquitination, which was detectable even in the absence of proteasome or lysosome inhibitors, the ΔRING mutant failed to induce detectable upregulation of EGFR ubiquitination even in the presence of both proteasome and lysosome inhibitors (MG132 and BAF, respectively) (*Figure 4F*). Together, these results demonstrate that ZNRF3 and RNF43 regulate EGFR ubiquitination and degradation through their E3 ligase activity.

## ZNRF3 and RNF43 form a complex with EGFR through the ECD

We next investigated whether ZNRF3 and RNF43 interact with EGFR to regulate its ubiquitination. In a co-IP experiment using lysates from MDA-MB-231 cells co-infected with lentivirus carrying EGFR and Myc-tagged ZNRF3/RNF43, anti-EGFR IP pulled down overexpressed ZNRF3/RNF43 (*Figure 5A*), indicating a complex between EGFR and ZNRF3/RNF43. Mass spectrometry of anti-EGFR immunoprecipitates identified endogenous ZNRF3 protein (*Figure 5—figure supplement 1A*), confirming ZNRF3 as an interacting partner of EGFR. Importantly, a proximity ligation assay (PLA) provided visual confirmation of the close interaction between co-transfected ZNRF3 and EGFR in situ (*Figure 5B*). Next, we sought to determine the specific regions involved in the interaction between ZNRF3/RNF43 and EGFR. ZNRF3 and RNF43 are transmembrane proteins comprising a long ECD, a single-span transmembrane domain (TM), and a catalytic intracellular domain (ICD) (*Figure 5C*). We thus generated constructs expressing ZNRF3 ECD-TM and TM-ICD, with an N-terminal 3× Myc-tag inserted after the signal peptide for detection (*Figure 5C*). Immunofluorescence (IF) confirmed the membrane localization of these two peptides (*Figure 5D*), after which we performed co-IP using

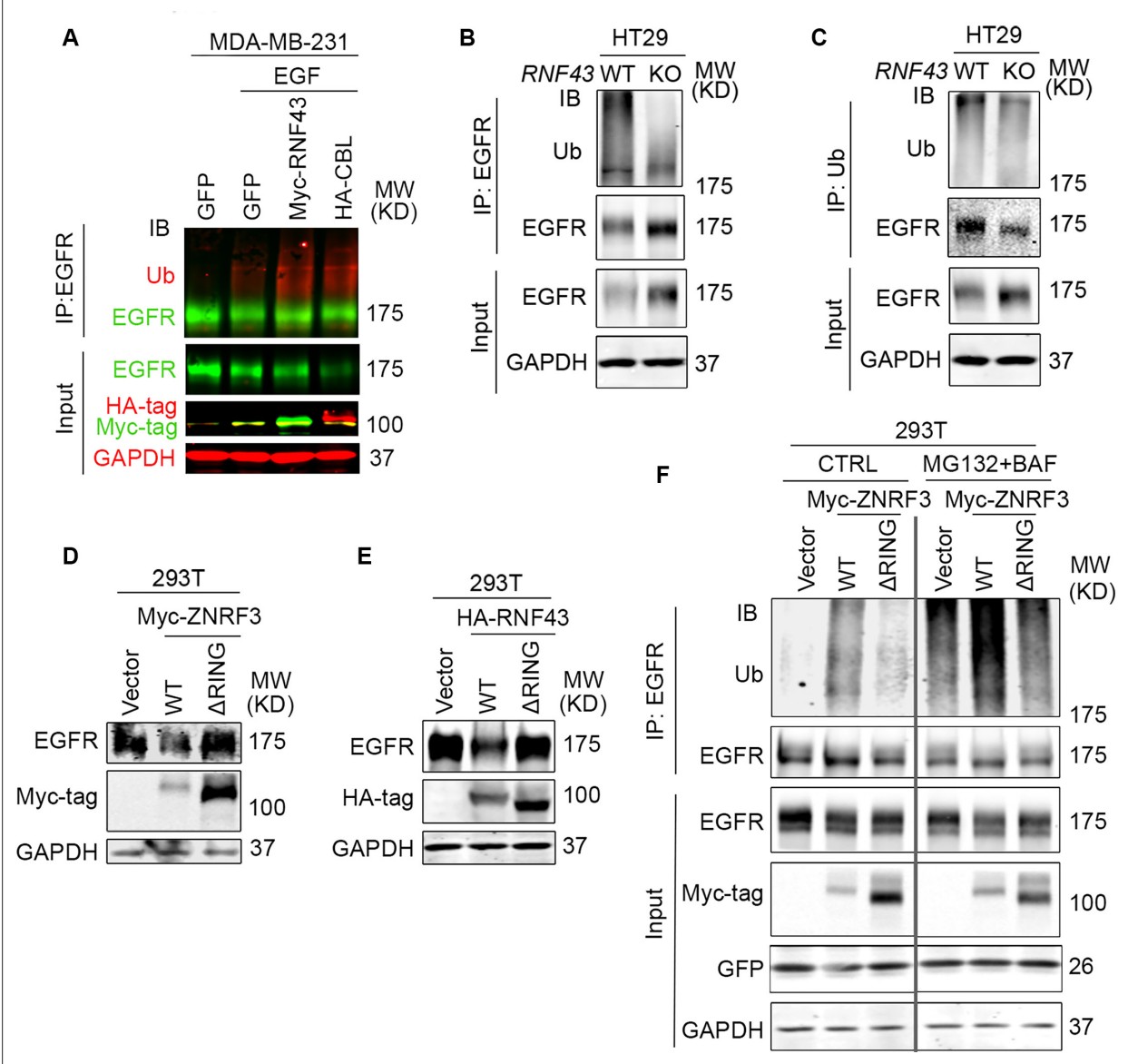

**Figure 4.** ZNRF3/RNF43 enhances EGFR ubiquitination through the RING domain. (**A**) Overexpression of RNF43 enhances EGFR ubiquitination level upon EGF (50 ng/ml) stimulation in MDA-MB-231 cells. CBL serves as a positive control. Cells were co-infected with lentivirus expressing EGFR and GFP, RNF43, or CBL. (**B, C**) Knockout of *RNF43* decreases EGFR ubiquitination in HT29 cells. (**B**) EGFR ubiquitination was examined by ubiquitin (Ub) immunoprecipitation (IP) followed by EGFR immunoblotting (IB). (**C**) HT29 cells were pretreated with 20 μM MG132 and 100 nM Bafilomycin A1 for 4 hr. EGFR ubiquitination after EGF treatment (50 ng/ml, 30 min) was examined by EGFR IP followed by Ub IB. (**D, E**) ZNRF3/RNF43 downregulates EGFR protein level through the RING domain. 293T cells were co-transfected with EGFR and vector, ZNRF3 WT or ΔRING mutant (**D**), RNF43 WT or ΔRING mutant (**E**). (**F**) ZNRF3 regulates EGFR ubiquitination through the RING domain. 293T cells were co-transfected with EGFR and vector, ZNRF3 WT or ΔRING mutant. EGFR ubiquitination after EGF treatment (50 ng/ml, 10 min) was examined by EGFR IP followed by Ub IB. Cells were pretreated with 20 μM MG132 and 100 nM Bafilomycin A1 for 4 hr.

The online version of this article includes the following source data and figure supplement(s) for figure 4:

**Source data 1.** PDF files containing the original, labeled blots and gels to *Figure 4*.

**Source data 2.** TIF files of the raw blots and gels to *Figure 4*.

**Figure supplement 1.** RNF43 loss decreases EGFR ubiquitination.

**Figure supplement 1—source data 1.** Excel file providing the numerical source data to *Figure 4—figure supplement 1*.

**Figure supplement 1—source data 2.** PDF files containing the original, labeled blots and gels to *Figure 4—figure supplement 1*.

**Figure supplement 1—source data 3.** TIF files of the raw blots and gels to *Figure 4—figure supplement 1*.

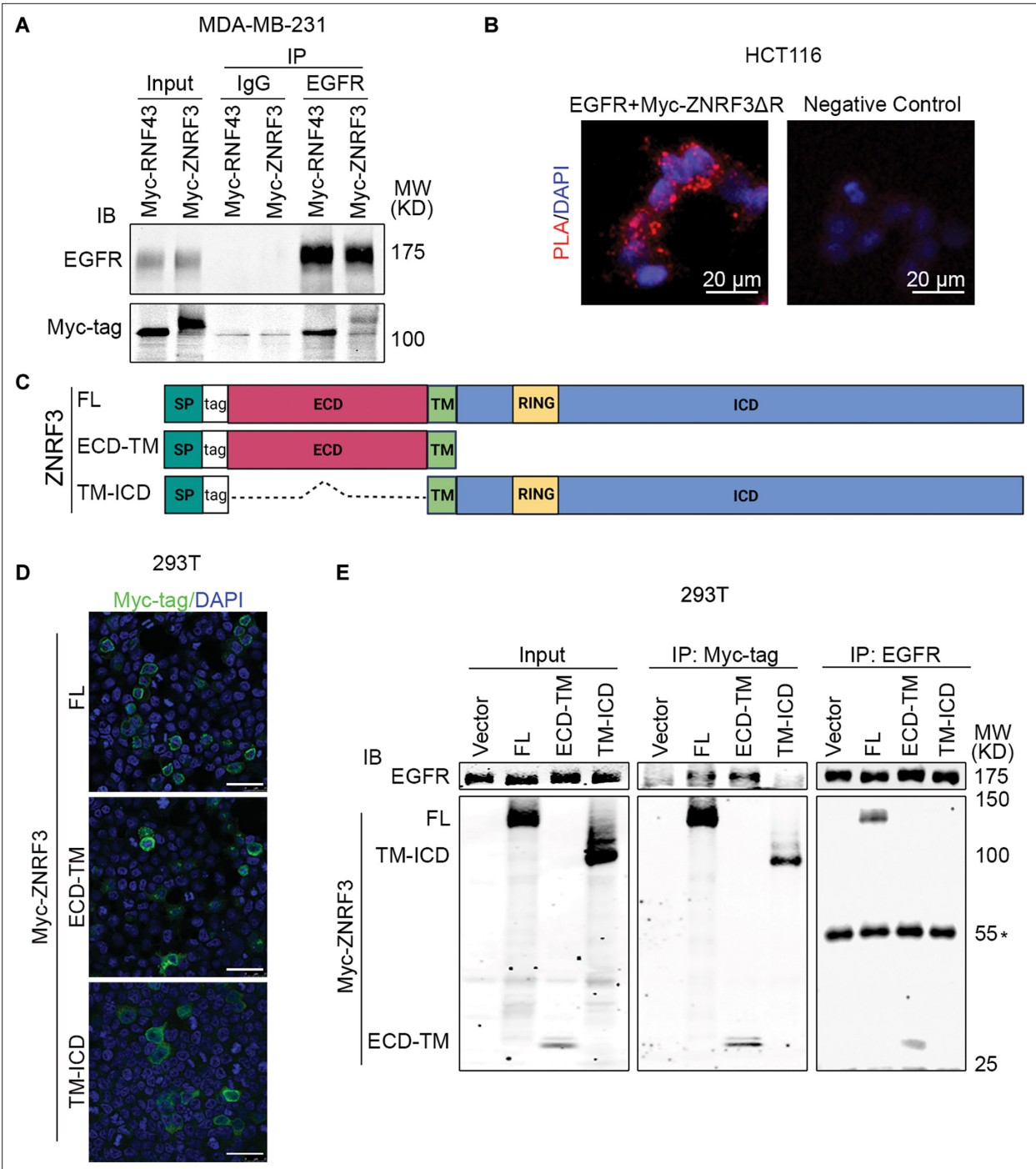

**Figure 5.** ZNRF3/RNF43 interacts with EGFR through the extracellular domain. (**A**) Ectopically expressed EGFR is co-immunoprecipitated with Myc-tagged RNF43 and ZNRF3 in MDA-MB-231 cells. (**B**) Representative images of proximity ligation assay (PLA) in HCT116 cells co-transfected with EGFR and Myc-ZNRF3ΔRING. Red, PLA signals; blue, DAPI nuclei staining; scale bar = 20 μm. (**C**) Schematic diagram of tagged ZNRF3 proteins. SP, signal peptide; FL, full-length; ECD, extracellular domain; TM, transmembrane domain; ICD, intracellular domain; RING, E3 ligase RING domain. (**D**) Immunofluorescence staining for ZNRF3 in 293T cells expressing Myc-tagged ZNRF3 FL, ECD-TM, TM-ICD. Scale bar = 40 μm. (**E**) ZNRF3 extracellular domain is required for ZNRF3 interaction with EGFR. 293T cells were co-transfected with EGFR and Myc-tagged ZNRF3 constructs, and the lysate amounts were adjusted to achieve comparable levels of EGFR protein in each immunoprecipitation (IP) system (input, left panel). EGFR interaction with ZNRF3 FL, ECD-TM, or TM-ICD was examined by Myc-tag IP followed by EGFR immunoblotting (IB) (middle panel) or by EGFR IP followed by Myc-tag IB (right panel). *, IgG heavy chain.

The online version of this article includes the following source data and figure supplement(s) for figure 5:

*Figure 5 continued on next page*

Figure 5 continued

**Source data 1.** PDF files containing the original, labeled blots and gels to *Figure 5*.

**Source data 2.** TIF files of the raw blots and gels to *Figure 5*.

**Figure supplement 1.** ZNRF3/RNF43 interacts with EGFR.

**Figure supplement 1—source data 1.** PDF files containing the original, labeled blots and gels to *Figure 5—figure supplement 1*.

**Figure supplement 1—source data 2.** TIF files of the raw blots and gels to *Figure 4—figure supplement 1*.

**Figure supplement 2.** The protease associate domain of ZNRF3/RNF43 is dispensable for EGFR interaction.

**Figure supplement 2—source data 1.** Excel file providing the numerical source data to *Figure 5—figure supplement 2*.

**Figure supplement 2—source data 2.** PDF files containing the original, labeled blots and gels to *Figure 5—figure supplement 2*.

**Figure supplement 2—source data 3.** TIF files of the raw blots and gels to *Figure 5—figure supplement 2*.

anti-Myc. EGFR was co-immunoprecipitated with Myc-tagged ZNRF3 full-length (FL) and ECD-TM, but not TM-ICD (*Figure 5E*). Conversely, anti-EGFR IP pulled down ZNRF3 FL and ECD-TM, but not TM-ICD (*Figure 5E*). Similar results were obtained when performing anti-HA IP of HA-tagged RNF43 FL and ECD-TM, which also brought down EGFR (*Figure 5—figure supplement 1B, C*). These data collectively indicate that ZNRF3 and RNF43 interact with EGFR via their ECDs, in accordance with evidence indicating the ZNRF3/RNF43 interaction with Frizzled via ECDs (*Tsukiyama et al., 2015*). This is in contrast with reports detecting ZNRF3/RN43 interaction with Frizzled via their ICDs and other adaptor proteins (*Jiang et al., 2015*).

The protease-associated (PA) domain is the only part conserved between the extracellular fragments of ZNRF3 and RNF43 (*Tsukiyama et al., 2021*). Therefore, we tested whether this domain is necessary for the interaction between ZNRF3/RNF43 with EGFR. We created PA domain-deletion mutants of ZNRF3/RNF43 (*Figure 5—figure supplement 2A*) and found that these deletion mutants (ΔPA) also interacted with EGFR (*Figure 5—figure supplement 2B and C*), indicating that the PA domain is dispensable for ZNRF3/RNF43 interaction with EGFR. In accordance with previous reports that the PA domain is dispensable for suppression of WNT signaling (*Jiang et al., 2015*; *Radaszkiewicz and Bryja, 2020*), we found that ΔPA ZNRF3 retained the ability to suppress the Top-Flash WNT reporter (*Figure 5—figure supplement 2D*).

## Loss of ZNRF3 and RNF43 unleashes EGFR-mediated cell growth in 2D culture and organoids

EGFR is an important tyrosine kinase that mediates many cellular activities, especially cell growth (*Wieduwilt and Moasser, 2008*; *Yue et al., 2021*). To study the biological consequence of ZNRF3/RNF43 downregulating EGFR, we first compared cell growth of WT and *ZNRF3/RNF43*-depleted cells. *Znrf3* KO in MEFs led to the upregulation of EGFR (*Figure 2B*) and increased cell growth (*Figure 6A*), without affecting canonical WNT signaling activity based on qPCR for WNT target genes (*Sox2*, *Axin2*, *Ccnd1*) (*Figure 6B*). EGF and EGFR signaling are also critical for the culture and maintenance of organoids derived from *Apc*-deficient mouse intestinal adenomas (*Sato et al., 2011*), which exhibit constitutively canonical WNT signaling that is no longer modulated by the RSPO-ZNRF3/RNF43 control at the membrane receptor level (*Tsukiyama et al., 2020*; *Tsukiyama et al., 2021*). Therefore, we established intestinal tumor organoids using the intestinal polyps from the *Apc*^min mice (*Evans et al., 1992*) and supplemented RSPO1 in the organoid culture medium to investigate the effect of ZNRF3/RNF43 deactivation on organoid growth and EGFR. Remarkably, the addition of RSPO1 increased the size of *Apc*^min mouse intestinal tumor organoids (*Figure 6C and D*) in agreement with a previous report (*Lähde et al., 2021*) although it did not significantly affect the frequency of organoids detected (*Figure 6E*). IB analysis confirmed elevated EGFR levels after RSPO1 treatment (*Figure 6F*) while qPCR showed no significant increase of *Egfr* nor WNT target genes, including *Axin2*, *Myc*, *Sox2*, and *Cd44* (*Figure 6G*). Furthermore, we performed functional assays in HT29 cells, which contain a mutated *APC*. Overexpression of ZNRF3 in HT29 substantially inhibited cell growth (*Figure 6H*) and reduced EGFR levels (*Figure 6I*). Conversely, KO of *RNF43* in HT29 activated multiple EGFR downstream effectors, as RPPA detected increased levels of P-AKT, P-ERK, P-GSK3, P-STAT1, and P-PRAS40 (*Figure 6—figure supplement 1*). Importantly, treatment with the EGFR activity inhibitor erlotinib, which abolished EGFR phosphorylation (*Figure 6J*), blocked the growth gain caused by

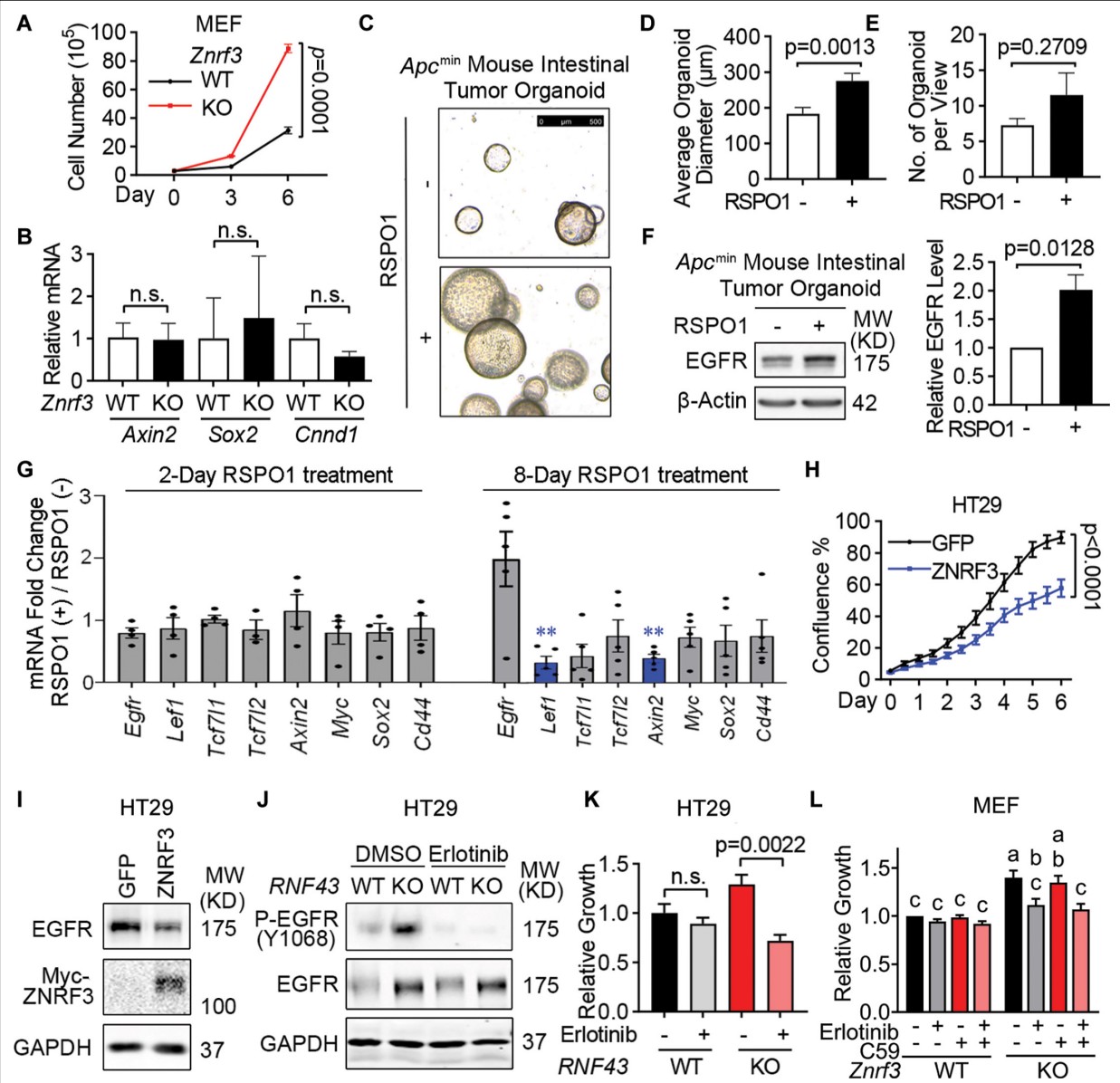

**Figure 6.** ZNRF3/RNF43 inhibits EGFR-mediated cell growth. (**A**) Knockout (KO) of *Znrf3* enhances murine embryonic fibroblast (MEF) cell growth, as measured by cell counting at the indicated time points. p-Values were calculated by two-way ANOVA uncorrected Fisher's LSD test. (**B**) Quantitative PCR (qPCR) analysis for WNT target genes in WT and *Znrf3* KO MEFs. Means ± SEMs are shown for this and other graphs. p-Values were calculated by Welch's t-test. n.s., not significant. (**C–E**) Supplementing RSPO1 promotes *Apc*^min^ mouse intestinal tumor organoid growth. The equal number of single cells from *Apc*^min^ mouse intestinal tumor organoids was embedded in Matrigel and cultured without or with 10% RSPO1 conditioned medium for 8 days. Representative images (**C**), and quantification of the size (**D**) and number (**E**) of formed *Apc*^min^ mouse intestinal tumor organoids are shown. Scale bar = 500 μm. p-Values were calculated by Welch's t-test. (**F**) Supplementing RSPO1 enhances EGFR protein level in *Apc*^min^ mouse intestinal tumor organoids. Representative images (left panel) and quantification (right panel) of EGFR protein level are shown. p-Values were calculated by Welch's t-test. (**G**) qPCR analysis for *Egfr* and WNT target genes in *Apc*^min^ mouse intestinal tumor organoids cultured with or without RSPO1 supplements. Genes with no significant changes after RSPO1 treatment were plotted in gray, genes significantly downregulated after RSPO1 treatment were plotted in blue. p-Values were calculated by Welch's t-test. **, p-value<0.01. (**H**) Overexpression of ZNRF3 inhibits HT29 cell growth. HT29 cells stably overexpressing GFP or ZNRF3 were seeded in equal numbers and measured by confluence percentage using Incucyte. p-Values were calculated by two-way ANOVA uncorrected Fisher's LSD test. (**I**) Overexpression of ZNRF3 reduces EGFR protein level in HT29 cells. (**J**) Erlotinib treatment blocks EGFR phosphorylation in WT and *RNF43* KO HT29 cells. Cells were treated with 5 μM erlotinib for 48 hr. (**K**) Erlotinib treatment inhibits cell growth in *RNF43* KO HT29 cells. p-Values were calculated by one-way ANOVA uncorrected Fisher's LSD test. (**L**) Erlotinib treatment inhibits cell growth in *Znrf3* KO MEF cells. *Znrf3* KO and WT MEF cells were treated with 0.5 μM erlotinib, or 0.1 μM WNT-C59 or both for 96 hr, and the growth was determined by CCK-8 assay. Means ± SEMs are shown. p-Values were calculated by two-way ANOVA, and the significance is presented by compact letter display. Columns marked by the same letter exhibit significant differences.

*Figure 6 continued on next page*

*Figure 6 continued*

The online version of this article includes the following source data and figure supplement(s) for figure 6:

**Source data 1.** Excel file providing the numerical source data to *Figure 6*.

**Source data 2.** PDF files containing the original, labeled blots and gels to *Figure 6*.

**Source data 3.** TIF files of the raw blots and gels to *Figure 6*.

**Figure supplement 1.** Reverse Phase Protein Array (RPPA) identifies EGFR downstream signaling molecules upregulated by *RNF43* knockout in HT29 cells.

**Figure supplement 1—source data 1.** Excel file providing the numerical source data to *Figure 6—figure supplement 1*.

**Figure supplement 2.** Quantitative PCR (qPCR) analysis for TGF-β signaling relevant genes in *Apc*min mouse intestinal tumor organoids cultured with or without RSPO1 supplements.

**Figure supplement 2—source data 1.** Excel file providing the numerical source data to *Figure 6—figure supplement 2*.

*RNF43* KO (*Figure 6K*). We also treated WT and *Znrf3* KO MEF cells with erlotinib, the WNT inhibitor Wnt-C59 (a small molecule that suppresses porcupine, which is required for palmitoylation and secretion of Wnt; *Proffitt et al., 2013*), or their combination. Neither inhibitor impacted WT MEF cell growth. In contrast, erlotinib, but not WNT-C59, significantly blocked the growth gain caused by *Znrf3* KO (*Figure 6L*), suggesting *Znrf3* KO induced MEF growth primarily via EGFR signaling rather than WNT signaling. Together, these data indicate that loss of *ZNRF3* and *RNF43* can promote tumor cell growth through EGFR signaling, sometimes even without substantially engaging canonical WNT signaling.

## ZNRF3/RNF43 inactivation enhances EGFR and promotes tumor growth in vivo

To demonstrate the function of the ZNRF3/RNF43-EGFR signaling pathway in vivo, we first xenografted luciferase-labeled HT29 cells overexpressing either GFP or ZNRF3 into NSG mice by flank injection and monitored tumor growth by bioluminescence imaging (*Figure 7A*). We found that overexpression of ZNRF3 significantly inhibited tumor growth (*Figure 7A*). These tumors exhibited decreased EGFR and P-EGFR levels compared to WT tumors (*Figure 7B*). However, canonical WNT signaling, based on levels of active-β-catenin (non-phosphorylated at Ser33/37/Thr41; *Figure 7B*), remained unaffected, suggesting that ZNRF3 suppresses tumor growth through downregulation of EGRF signaling without substantial involvement of canonical WNT signaling.

Furthermore, we validated ZNRF3/RNF43-EGFR signaling in prostate cancer, one of the several other tumors in patients besides CRC that showed a negative correlation between *ZNRF3/RNF43* mRNA and EGFR protein levels (*Figure 1F*). Specifically, we tested whether ZNRF3/RNF43 loss affects EGFR signaling and promotes prostate tumorigenesis in vivo. By breeding mice that contain cre-inactivatable alleles of *Znrf3* and *Rnf43* (*Koo et al., 2012*) with mice transgenic for Probasin-Cre (*Wu et al., 2001*), we generated prostate-specific *Znrf3/Rnf43* KO mice. At 1 year of age, these mice exhibited multi-focal prostatic dysplasia and multifocal invasive tumors, as shown by the H&E images (*Figure 7C*) and pathological scores (*Figure 7D*, *Figure 7—figure supplement 1A–D*). Immunohistochemical staining revealed elevated levels of both EGFR and P-EGFR in *Znrf3/Rnf43* KO prostatic dysplasia and tumors, compared to WT prostate tissues (*Figure 7E*), indicating EGFR signaling activation as a result of *Znrf3/Rnf43* KO. We also observed more intense β-catenin staining in KO tissues and tumors than in WT mice, although the β-catenin signals were largely excluded from the nucleus in both WT and *Znrf3/Rnf43* KO mouse prostate tissues and tumors (*Figure 7E*). Together, these results suggest that loss of *ZNRF/RNF43* elevates EGFR levels and signaling, promoting tumor development, sometimes in conjunction with Wnt signaling activation and other times independent of Wnt signaling.

## Discussion

The two homologous E3 Ub ligases, ZNRF3 and RNF43, and their antagonistic ligand RSPO have been extensively studied as WNT signaling modulators during normal development and diseases (*Hao et al., 2012*; *Koo et al., 2012*; *Planas-Paz et al., 2016*; *Szenker-Ravi et al., 2018*; *Basham et al., 2019*; *Lee et al., 2020*; *Sun et al., 2021*). Their primary role has been attributed to negatively regulating Frizzled receptors through ubiquitination and degradation (*Hao et al., 2012*; *Koo et al.,*

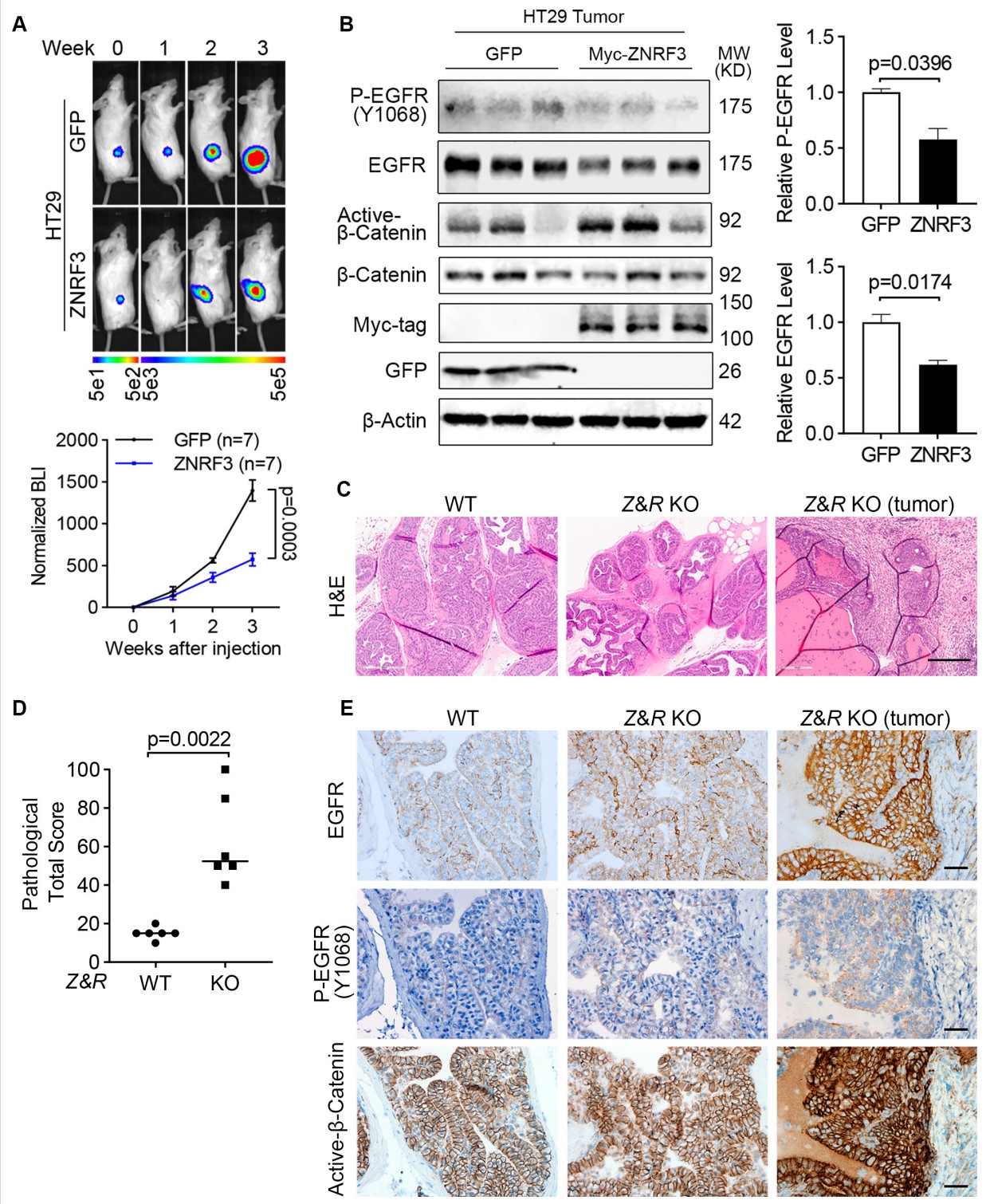

**Figure 7.** ZNRF3/RNF43 loss enhances EGFR signaling and promotes tumorigenesis. (**A**) Overexpression of ZNRF3 suppresses HT29 tumor growth in vivo. Representative bioluminescence images (top panel) and quantification (bottom panel) of flank-injected HT29 cells expressing either GFP or Myc-tagged ZNRF3. p-Values were calculated by two-way ANOVA uncorrected Fisher's LSD test. (**B**) Overexpression of ZNRF3 inhibits P-EGFR and total EGFR levels in HT29 tumors. Representative images (left panel) and quantification of P-EGFR (right-top panel) and total EGFR (right-bottom panel) protein levels are shown. Means ± SEMs are shown. p-Values were calculated by Welch's t-test (**B**). (**C**) Representative H&E images of prostate tissues from wild-type (WT) or prostate-specific *Znrf3/Rnf43* knockout mice. Prostate tissue or tumor samples were collected at 1 year of age. Scale bar = 300 μm. (**D**) Total pathological scores of prostate tissues from WT or prostate-specific *Znrf3/Rnf43* knockout mice. n=6 mice per group. p-Values were

*Figure 7 continued on next page*

Figure 7 continued

calculated by Mann-Whitney. (E) Representative images of immunochemistry staining for EGFR, P-EGFR, and active-β-catenin in prostate tissues from WT or prostate-specific *Znrf3*/*Rnf43* knockout mice. Scale bar = 40 µm.

The online version of this article includes the following source data and figure supplement(s) for figure 7:

**Source data 1.** Excel file providing the numerical source data to *Figure 7*.

**Source data 2.** PDF files containing the original, labeled blots and gels to *Figure 7*.

**Source data 3.** TIF files of the raw blots and gels to *Figure 7*.

**Figure supplement 1.** Pathological assessment on eccentric thickened wall (**A**), dysplasia (**B**), micro-invasion (**C**), and frank invasion (**D**) of wild-type (WT) and *Znrf3*/*Rnf43* knockout (KO) mouse prostate tissues.

**Figure supplement 1—source data 1.** Excel file providing the numerical source data to *Figure 7—figure supplement 1*.

*2012*), and the early mass spectrometry analysis of 293T cells following inducible expression of RNF43 only detected downregulation of Frizzled and LRP5 (*Koo et al., 2012*). However, subsequent reports suggested that ZNRF3/RNF43 might also regulate other substrates and exert distinct functions in context-dependent manners. For instance, ZNRF3/RNF43 may induce degradation of BMP receptor BMPR1A (*Lee et al., 2020*), maternal dorsal determinant Huluwa (*Zhu et al., 2021*), non-canonical WNT component VANGL2 (*Radaszkiewicz et al., 2021*), and the cell adhesion protein E-cadherin (*Zhang et al., 2019*) in different physiological or pathological conditions. In the context of cancer, *ZNRF3* and *RNF43* are frequently inactivated by gene deletion, mutation, or other means (*Seshagiri et al., 2012*; *Hao et al., 2016*). Therefore, our study focused on elucidating ZNRF3/RNF43-regulated signaling pathways in cancer. We discovered that EGFR is the protein most negatively associated with *ZNRF3*/*RNF43* expression using proteogenomic data of cancer patients, and that ZNRF3/RNF43 interacts with EGFR and downregulates EGFR through ubiquitination and degradation. Using multiple cell lines and animal models, we demonstrated that loss-of-function changes of ZNRF3/RNF43 elevate EGFR signaling activity to promote cancer progression.

In CRC, *ZNRF3* and *RNF43* mutations occur frequently (*Giannakis et al., 2014*; *Bond et al., 2016*; *Tu et al., 2019*; *Vasaikar et al., 2019*) and have been mainly associated with the potentiation of WNT signaling (*Hao et al., 2012*; *Koo et al., 2012*; *Hao et al., 2016*). However, our findings suggest that ZNRF3/RNF43 deactivation may unleash other key oncogenic pathways in CRC, especially in the context of APC/CTNNB1 co-mutations that bypass the need for WNT receptor activation for pathway activation (*Cerami et al., 2012*; *Gao et al., 2013*). We provide evidence that in human CRC cells with a hyperactive WNT pathway due to *APC* mutations, KO of *RNF43* promotes cancer cell growth by activating EGFR signaling. Additionally, we found that the *RNF43* G659Vfs*41 mutant, a hotspot frameshift mutation in human colorectal tumors that may not potentiate WNT signaling (*Tu et al., 2019*; *Li et al., 2020*), elevates EGFR protein levels, suggesting that this mutation may exert its onco-genic role in cancer through EGFR upregulation. Our data also have clinical implications, as germline polymorphisms and tumor gene expression levels of ZNRF3/RNF43 and RSPO may be related to tumor response to anti-EGFR monoclonal antibody cetuximab-based treatment in CRC patients (*Batt-aglin et al., 2020*; *Elez et al., 2022*), providing insights into understanding the interplay between RSPO-ZNRF3/RNF43 and EGFR protein levels and anti-EGFR activity in CRC treatment.

By identifying ZNRF3/RNF43 as crucial regulators linking EGFR and WNT signaling at the membrane receptor level, our work reveals a novel mechanism of EGFR and WNT co-activation, which is commonly observed in human cancers (*Hu and Li, 2010*). Further work should explore the extent to which EGFR and WNT receptors (Frizzled and LRP5/6) each transmit the effect of ZNRF3/RNF43 deactivation at different cancer stages and investigate the efficacy of blocking both EGFR and WNT signaling in the treatment of ZNRF3/RNF43-deactivated tumors. Our findings also have implications for organoid culture and normal development. For example, while RSPO and EGF are both required for organoid growth and maintenance, our data suggest that RSPO can both help sustain surface EGFR levels by deactivating ZNRF3/RNF43 and synergize with EGF to boost EGFR signaling activity. Comparative studies on ZNRF3/RNF43 regulation of EGFR versus WNT receptors in terms of substrate abundance, enzyme-substrate binding affinity, in vivo ligand availability, and temporal and spatial regulatory mech-anisms can provide a comprehensive understanding of the RSPO-ZNRF3/RNF43-regulated signaling network. Lahde et al. reported that RSPO1 addition to *Apc*-deficient mouse intestinal organoids led to an increased size but a decreased number, and their studies implicated abnormal recruitment of

the TGF-β/SMAD pathway via LGR5/TGFBRII heterodimers as a mechanism (*Lähde et al., 2021*). We did not detect enhanced TGF-β signaling in our *Apc*^min mouse intestinal tumor culture treated by RSPO1 for 2 or 8 days (*Figure 6—figure supplement 2*). Furthermore, we did not observe an impact of ZNRF3/RNF43 overexpression on TGFβ receptor I protein levels in our 293T culture studies (*Figure 2—figure supplement 1B*), although our bioinformatic analysis of patient CRCs revealed that SMAD4, a crucial transcription factor downstream of TGF-β receptors, was among the proteins down-regulated by ZRNF3/RNF43 (*Figure 1C*). Therefore, whether TGFβ signaling is directly affected by the RSPO-ZNRF3/RNF43-regulated signaling network needs to be further investigated.

Structural studies have identified the key amino acids and motifs mediating ZNRF3/RNF43 binding with their peptide ligand RSPO (*Chen et al., 2013*; *Peng et al., 2013*; *Zebisch et al., 2013*) and also suggested that RSPO has the closest structural homology with EGFR (domain IV) and insulin receptor (domain II) (*Chen et al., 2013*). Our co-IP data indicate that ZNRF3/RNF43 interact with EGFR through their ECDs. It would also be interesting to determine further the specific domains and amino acids that are responsible for ZNRF3/RNF43 and EGFR interaction and to study whether clinically relevant point mutations at the ECDs of ZNRF3/RNF43 and/or EGFR hinder their binding affinity, thereby enhancing EGFR signaling to promote cancer initiation and progression.

RSPO1-4 bind to both ZNRF3/RNF43 and LGR4/5/6, resulting in ZNRF3/RNF43 auto-ubiquitination and degradation (*Hao et al., 2012*; *Koo et al., 2012*). RSPO2/3 can also inhibit ZNRF3/RNF43 inde-pendently of LGR4/5/6 (*Lebensohn and Rohatgi, 2018*; *Park et al., 2018*; *Szenker-Ravi et al., 2018*) because RSPO2/3 can bind to RNF43/ZNRF3 with relatively high affinity, especially in the presence of heparan sulfate proteoglycans (HSPGs) (*Dubey et al., 2020*). Approximately 18% of CRCs harbor amplification or mutations of *RSPO1-4* or *LGR4/5/6*, while 10% of CRCs have recurrent *RSPO2/3* gene fusions (*Seshagiri et al., 2012*). These genetic alterations potentially lead to enhanced inhibition of ZNRF3/RNF43, activating WNT and EGFR signaling (*Seeber et al., 2019*). While the involvement of each co-receptor (LGR4/5/6; HSPGs) in regulating RSPO-ZNRF3/RNF43 degradation of EGFR requires more investigation, LGR4/5/6 may be dispensable for RSPO2-ZNRF3/RNF43 regulation of EGFR because we found that RSPO2 mutant that cannot bind to LGR4/5/6 still enhanced EGFR protein levels. It is worth noting that LGR4/5/6 may have a parallel mechanism controlling EGFR levels as we have previously observed that LGR4 can enhance EGFR signaling independently of WNT activation (*Yue et al., 2021*).

Our work also uncovers a new mechanism of regulating EGFR levels in cancer. Hyperactivation of EGFR in cancer has been attributed to activating mutations, gene amplification, aberrant gene expression, and defective endocytosis/degradation (*Nakai et al., 2016*). However, these mechanisms cannot account for all the 60–80% of CRC cases whose EGFR levels are upregulated (*Cohen, 2003*). ZNRF3/RNF43 loss-mediated EGFR stabilization represents a novel mechanism of EGFR upregula-tion, explaining the elevated EGFR protein levels observed in many human cancers without EGFR gene amplification or overexpression. In our study, ZNRF3/RNF43 can regulate EGFR levels under basal culture conditions, as well as with EGF ligand stimulation, but the commonly known EGFR E3 ligase, CBL, mediates ligand-dependent EGFR ubiquitination and degradation (*Batzer et al., 1994*; *Levkowitz et al., 1998*; *Levkowitz et al., 1999*; *Waterman et al., 1999*; *Waterman et al., 2002*; *Grøvdal et al., 2004*; *Duan et al., 2011*). Therefore, future studies should investigate whether ZNRF3/RNF43 and CBL regulate EGFR ubiquitination and degradation differently, whether ZNRF3/RNF43 regulate both ligand-independent and ligand-dependent EGFR ubiquitination/degradation and consequently affecting different downstream signaling pathways, and whether different EGFR ligands or ligand concentrations direct EGFR to ZNRF3/RNF43 versus CBL for ubiquitination and degradation (*Harris et al., 2003*; *Roepstorff et al., 2009*). In addition, a feedback regulation often contributes to tight controls of signaling activation (*Chandarlapaty, 2012*). In accordance, ZNRF3/RNF43 regulation of WNT signaling is feedback-controlled by *ZNRF3/RNF43* as WNT target genes (*Hao et al., 2012*; *Koo et al., 2012*). Whether and how EGFR signaling may affect levels and activities of ZNRF3/RNF43 and their partners including RSPOs and LGRs as a feedback regulation remain to be understood.

In conclusion, our study unveils a novel ZNRF3/RNF43-EGFR signaling axis in cancer and provides critical insights into RSPO-ZNRF3/RNF43 signaling during cancer progression. Understanding how ZNRF3/RNF43 regulates EGFR and the crosstalk between EGFR and WNT receptors may offer poten-tial therapeutic targets for cancer treatment. Additionally, our findings have implications in organoid culture and normal development, highlighting the significance of RSPO-ZNRF3/RNF43 signaling in

various physiological processes. Further research on ZNRF3/RNF43 regulation of EGFR versus WNT receptors in different cancer stages can contribute to a comprehensive understanding of RSPO-ZNRF3/RNF43 signaling in cancer progression and provide valuable knowledge for potential therapeutic interventions.

## Materials and methods

### Animal studies

All experiments using mice were performed utilizing procedures approved by the Institutional Animal Care and Use Committees at Baylor College of Medicine and the Van Andel Institute. The *Rnf43*-flox; *Znrf3*-flox were obtained from the laboratory of Hans Clevers (*Koo et al., 2012*). CMV-Cre animals were ordered from Jackson Laboratories (JAX stock #006054) (*Schwenk et al., 1995*). C57Bl/6J used for backcrossing were ordered from Jackson Laboratories (JAX stock #000664). Probasin-Cre animals were developed in the laboratory of Pradip Roy-Burman (*Wu et al., 2001*). NOD.Cg-Prkdc scid Il2rg tm1Wjl/SzJ (NSG) mice were purchased from Jackson Laboratories (JAX stock #005557) and bred in-house.

### Cell lines

MDA-MB-231, MDA-MB-468, and HEK293T cells were purchased from ATCC. HT29 and LS180 cells were provided by Q Liu; HCT116 cells were provided by N Shroyer; MEF cells were generated from e12.5 embryos using standard methods (*Herrera et al., 1996*). MDA-MB-231, MDA-MB-468, HT29, HCT116, HEK293T, and MEF cells were cultured in DMEM (Corning, 10-013-CV) with 10% FBS and 1% penicillin/streptomycin. LS180 cells were cultured in RPMI1640 (Corning, 10-013-CV) with 10% FBS and 1% penicillin/streptomycin. All cell lines were routinely tested for mycoplasma contamination.

### Constructs

Plasmids expressing Flag-FZD5, RNF43, HA-RNF43, or HA-RNF43 (CD8 sp) were provided by Q Liu. Plasmids expressing Myc-RNF43, Myc-ZNRF3, or Myc-ZNRF3 ΔRING were provided by F Cong. Plasmids for Myc-ZNRF3-ECD-TM, Myc-ZNRF3-TM-ICD, Myc-ZNRF3 ΔPA, HA-RNF43 ΔPA, HA-RNF43 ΔRING, HA-RNF43-ECD-TM were constructed by subcloning using In-Fusion (TaKaRa, 639642). Plasmids were further constructed for lentiviral expression by subcloning into pBobi vector by In-Fusion. Flag-EGFR (L858R) is PCR-amplified and subcloned into FUCGW vector at EcoRI site for lentiviral expression.

### Generation of *RNF43* or *ZNRF3* KO cancer cell lines

HT29 *RNF43* KO cells were provided by Q Liu (guide RNA sequence, g0: 5'-GGCTGCTGATGGCTAC CCTGC-3'). Other CRISPR guide sequences for knocking out *RNF43* were designed by http://crispr.mit.edu and cloned into lentiCRISPR v2 (Addgene 52961). Sequences were as follows: g1: 5'-TGGA CGCACAGGACTGGTAC-3'; g2: 5'-CAGAGTGATCCCCTTGAAAA-3'; g3: 5'-GGGCAGCCAGCT GCAGCTGG-3'. Lentiviral construct for knocking out *ZNRF3* was provided by Q Liu (guide RNA sequence, 5'- AGGACTTGTATGAATATGGC-3') (*Jiang et al., 2015*; *Tu et al., 2019*). Cancer cells were infected with lentivirus-carrying CRISPR constructs or vector, and then selected in the culture medium containing 2 µg/ml puromycin (InvivoGen, ant-pr-1).

### Generation of *Znrf3* KO MEFs

Global null alleles of *Rnf43* and *Znrf3* were generated in vivo by crossing *Rnf43*-flox; *Znrf3*-flox mice (*Koo et al., 2012*) with CMV-Cre (*Schwenk et al., 1995*). The resulting mice were backcrossed to C57Bl/6J animals to remove the Cre transgene and generate global deletion mouse colonies. Global deletions were confirmed using allele-specific PCR for both *Rnf43* and *Znrf3*. MEFs were generated using standard protocols (*Durkin et al., 2013*). Briefly, *Rnf43*^KO/+; *Znrf3*^KO/+ mice were crossed, and embryos were collected at E12.5. Yolk sacs were collected for genotyping, and embryo heads and internal organs were removed. The remaining tissue was minced using a razor blade and 0.5 ml 0.05% Trypsin-EDTA was added. The suspension was incubated at 37°C for 30 min, then cultured in MEF media (DMEM, 10% FBS, 1× PenStrep) for subsequent experiments.

## Chemicals

Erlotonib (Selleck Chemicals) was dissolved in ethanol, and Wnt-C59 (Cellagen Technology) was dissolved in DMSO.

## Cell proliferation assay

MEF cells were seeded at 5000 cells in 96-well and the media was refreshed with drugs the next day. On day 4 post-drug treatments, cells were incubated with Cell Counting Kit-8 (CCK-8) reagents (ApexBio) for 4 hr, and the growth was determined at 450 nm by spectrophotometer.

## $Apc^{min}$ mouse intestinal tumor organoid culture

Intestinal polyps from 6-month-old female $Apc^{min}$ mice (**Evans et al., 1992**) were isolated and maintained using previously described protocols (**Sato et al., 2011**; **Xue and Shah, 2013**), embedded in Matrigel (Corning, 356231), and cultured in basal culture medium (advanced Dulbecco's modified Eagle medium/F12 [Invitrogen, 12634-028], penicillin/streptomycin [Invitrogen, 15140-122], 10 mmol/l HEPES [Invitrogen, 15630-080], 2 mM GlutaMAX [Invitrogen, 35050-061], 1×N2 [Invitrogen, 17502-048], 1×B27 [Invitrogen, 17504-044], 1 mmol/l N-acetylcysteine [Sigma-Aldrich, A9165-5G]) supplemented with 50 ng/ml mouse recombinant EGF (Invitrogen, PMG8043). The organoids were trypsinized with 0.05% Trypsin-EDTA (Invitrogen, 25300054) to obtain single cells, counted, and embedded into Matrigel in equal numbers, cultured in basal culture medium supplemented with EGF or basal culture medium supplemented with EGF and 10% RSPO1 conditioned medium for 8 days. The organoids were imaged using a Leica DMi8 Inverted Microscope and then harvested for immunoblot analysis.

## RNA extraction and qPCR

Cells were lysed with TRIzol reagent (Invitrogen, 15596-026). Total RNA was extracted by chloroform and isopropanol precipitation. cDNA was obtained using the SuperScript III First-Strand Synthesis System (Invitrogen, 18080-051). qPCR analyses were performed with primers listed below using SsoAdvanced Universal SYBR Green Supermix (Bio-Rad, 1725270) with at least duplicate samples in three independent experiments. Plotted are data normalized to *ACTB* or *Actb* and relative to the control. The following primers were used: *ACTB*: 5'-ACTCTTCCAGCCTTCCTTCC-3', 5'-CAGTGATC TCCTTCTGCATCC-3'; *EGFR*: 5'-TGCCCATGAGAAATTTACAGG-3', 5'-ATGTTGCTGAGAAAGTCACT GC-3'; *Actb*: 5'- CATTGCTGACAGGATGCAGAAGG-3', 5'-TGCTGGAAGGTGGACAGTGAGG-3'; *Egfr*: 5'- GGACTGTGTCTCCTGCCAGAAT-3', 5'-GGCAGACATTCTGGATGGCACT-3'; *Axin2*: 5'-ATGG AGTCCCTCCTTACCGCAT-3', 5'-GTTCCACAGGCGTCATCTCCTT-3'; *Sox2*: 5'-AACGGCAGCTAC AGCATGATGC-3', 5'-CGAGCTGGTCATGGAGTTGTAC-3'; *Ccnd1*: 5'- GCAGAAGGAGATTGTGCCAT CC-3', 5'-AGGAAGCGGTCCAGGTAGTTCA-3'. The primer sequences for *Tgfbr1*, *Tgfbr2*, *Smad4*, *Ep300*, *Bach1* were from OriGene.

## Flow cytometric analysis

Cells were trypsinized and resuspended in antibody dilution buffer (0.5% BSA in PBS). $5×10^5$ cells were incubated with EGFR (D1D4J) XP Rabbit antibody (PE Conjugate) (Cell Signaling Technology, 48685) for 1 hr at 4°C. Cells were then washed with antibody dilution buffer and resuspended in antibody dilution buffer. The cells were then subjected to flow cytometry using a BD FACSCanto II (BD, NJ, USA). The resulting data were analyzed using the BD FACSDiva software.

## Immunofluorescence

HEK293T cells were seeded on poly-D-lysine-coated chamber slides (Corning, 354632) and transfected with empty vector, or constructs expressing WT or fragments of RNF43 or ZNRF3. 48 hr after transfection, cells were fixed in 3.2% paraformaldehyde in PBS at RT for 15 min, permeabilized in 3% BSA/0.1% saponin (Sigma-Aldrich, 47036) in PBS at RT for 30 min, and then incubated in the secondary antibody anti-Rabbit Alex 488 (Invitrogen, A32731) at RT for 1 hr. Afterward, the cells were stained with DAPI (Thermo Fisher Scientific, 62248) and mounted.

## Immunohistochemistry

Immunochemistry staining was performed using the Vectastain Elite ABC system (Vector Laboratories, Burlingame, CA, USA) and developed using DAB as chromogen (Agilent Dako, Santa Clara, CA, USA).

Primary antibodies used in the study included EGFR (D1P9C) rabbit antibody (Mouse Preferred, Cell Signaling Technology, 71655), P-EGFR (Y1068) XP rabbit antibody (Cell Signaling Technology, 2234), non-phospho (Active) β-catenin (Ser33/37/Thr41) (D13A1) rabbit antibody (Cell Signaling Technology, 8814).

### Proximity ligation assay

HCT116 cells were seeded onto poly-D-lysine-coated chamber slide (Corning) and co-transfected with Myc-ZNRF3ΔR and EGFR. Two days after transfection, the cells were processed following the standard protocol provided by Sigma-Aldrich (Duolink In Situ Red Starter Kit Mouse/Rabbit). The primary antibodies, EGFR (D38B1) XP rabbit antibody (Cell Signaling Technology, 4267) and Myc-tag (9B11) mouse antibody (Cell Signaling Technology, 2276), were used to detect ZNRF3 and EGFR inter-action. For negative control, no primary antibody was used. DAPI was used to stain nuclei. The stained cells were imaged using Leica DMi8 Fluorescence Microscope.

### Western blot

Cells were lysed on ice using RIPA lysis buffer supplemented with protease inhibitors (Sigma-Aldrich, P8340) and phosphatase inhibitors (Sigma-Aldrich, P5726 and P0044). Cell lysates were then centrifuged at 14,000 rpm for 15 min at 4°C. Supernatant was collected, and the concentration of protein lysate was quantified by BCA assay (Thermo Fisher Scientific, 23225). Cell lysates were mixed with Laemmli sample buffer (Bio-Rad, 1610747) and β-mercaptoethanol (Thermo Fisher Scientific, 21985023) before boiling for 5 min at 95°C. Equal amounts of protein lysates were loaded and run in 8–12% SDS-PAGE gel. Gels were transferred onto nitrocellulose membrane (Thermo Fisher Scientific, 88018) at 100 V for 90 min at 4°C. Membranes were then blocked with 5% non-fat milk in TBST at RT for 1 hr, incubated with primary antibodies overnight at 4°C and secondary antibodies at RT for 1 hr, and scanned using the Odyssey LI-COR imaging system. Primary antibodies used in the study included EGFR (D38B1) XP rabbit antibody (Cell Signaling Technology, 4267), EGFR (A-10) mouse antibody (Santa Cruz, sc-373746), EGFR (D1P9C) rabbit antibody (Mouse Preferred, Cell Signaling Technology, 71655), P-EGFR (Y1068) XP rabbit antibody (Cell Signaling Technology, 3777), HA-tag mouse antibody (BioLegend 901501), HA-tag (C29F4) rabbit antibody (Cell Signaling Technology, 3724), Myc-tag (9B11) mouse antibody (Cell Signaling Technology, 2276), Myc-tag (71D10) rabbit antibody (Cell Signaling Technology, 2278), Flag-tag M2 mouse antibody (Sigma-Aldrich, F1804), Ub (P4D1) mouse antibody (Santa Cruz, sc-8017), CBL mouse antibody (Santa Cruz, sc-1651), non-phospho (Active) β-catenin (Ser33/37/Thr41) (D13A1) rabbit antibody (Cell Signaling Technology, 8814), β-catenin mouse antibody (BD Biosciences, 610153), GAPDH rabbit antibody (Santa Cruz, sc-25778), β-actin (Sigma, A5441), and α-tubulin mouse antibody (Sigma-Aldrich, T9026). Secondary antibodies used included anti-Mouse IRDye 680RD (LI-COR, 926-68070) and anti-Rabbit IRDye 800CW (LI-COR, 926-32211).

### IP assay

Cells were lysed on ice using NP-40 or RIPA lysis buffer supplemented with protease inhibitors (Sigma-Aldrich, P8340). Cell lysates were then centrifuged at 14,000 rpm for 15 min at 4°C. Supernatant was collected, and the concentration of protein lysate was quantified by BCA. Protein lysates were incubated with either EGFR (Ab-13) mouse antibody (Thermo Fisher Scientific, MS-609-P1) or Ub (P4D1) mouse antibody (Santa Cruz, sc-8017), and Protein G Sepharose beads (GE Healthcare, 17061801) overnight at 4°C. Beads were washed with NP-40 or RIPA lysis buffer. Whole-cell lysates and immuno-precipitates were analyzed by western blot analysis.

### IP and mass spectrometry

The IP-MS experiments were performed as previously described (*Chen et al., 2019*). BCG823 cells were starved overnight and then treated with 50 ng/ml EGF for 120 min before sample collection. For each IP experiment, 1 mg protein lysate was incubated with 5 µg EGFR (Ab-13) antibody for 2 hr at 4°C and cleared by ultracentrifugation (100,000×*g*, 15 min). The supernatant was then incubated with 30 µl 50% protein A-Sepharose slurry (GE Healthcare, 17-0780-01) for 1 hr at 4°C. The bead-bound complexes were washed four times with NETN buffer (20 mM Tris pH 7.5, 1 mM EDTA, 0.5% NP-40, 150 mM NaCl) and eluted with 20 µl 2× Laemmli buffer and heated at 95°C for 10 min. IP samples

were resolved on a NuPAGE 10% Bis-Tris gel (Life Technologies, WG1201BX10) in 1× MOPS running buffer; the gel was cut into 3 molecular weight regions plus the IgG heavy and light chain bands. Each band was in-gel digested overnight with 100 ng of trypsin, which cleaves peptide chains at the C-termini of lysine or arginine in 20 µl of 50 mM $NH_4HCO_3$ at 37°C. Peptides were extracted with 350 µl of 100% acetonitrile and 20 µl of 2% formic acid, and dried in a Savant Speed-Vac. Digested peptides were dissolved in 10 µl of loading solution (5% methanol containing 0.1% formic acid) and subjected to LC-MS/MS assay as described previously (*Dharmat et al., 2018*).

## Reverse phase protein array

RPPA assays were carried out as described previously with minor modifications (*Creighton and Huang, 2015*). Protein lysates were prepared with modified TPER buffer supplemented with a cocktail of protease and phosphatase inhibitors. Protein lysates at 0.5 mg/ml of total protein were denatured in SDS sample buffer containing 2.5% β-mercaptoethanol at 100°C for 8 min. The Aushon 2470 Arrayer (Aushon BioSystems) with a 40 pin (185 µm) configuration was used to spot samples and control lysates onto nitrocellulose-coated slides (Grace Bio-labs) using an array format of 960 lysates/slide (2880 spots/slide). The slides were probed with a set of 216 antibodies against total and phosphoproteins using an automated slide stainer Autolink 48 (Dako). Each slide was incubated with one specific primary antibody, and a negative control slide was incubated with antibody diluent instead of primary antibody. Primary antibody binding was detected using a biotinylated secondary antibody followed by streptavidin-conjugated IRDye680 fluorophore (LI-COR Biosciences). Total protein content of each spotted lysate was assessed by fluorescent staining with Sypro Ruby Protein Blot Stain according to the manufacturer's instructions (Molecular Probes). Fluorescence-labeled slides were scanned on a GenePix 4400 AL scanner, along with accompanying negative control slides, at an appropriate PMT to obtain optimal signal for this specific set of samples. The images were analyzed with GenePix Pro 7.0 (Molecular Devices). Total fluorescence signal intensities of each spot were obtained after subtracting the local background signal for each slide and were then normalized for variation in total protein, background, and non-specific labeling using a group-based normalization method. For each spot on the array, the background-subtracted foreground signal intensity was subtracted by the corresponding signal intensity of the negative control slide (omission of primary antibody) and then normalized to the corresponding signal intensity of total protein for that spot. Each image and its normalized data were evaluated for quality through manual inspection and control samples. Antibody slides that failed the quality inspection were either repeated at the end of the staining runs or removed before data reporting. Samples (in four biological replicates) were extracted from the normalized data and then log2 transformed. The median value of the three technical replicates was used for statistical analysis. Welch's t-tests were used for group comparisons. FDR-adjusted p-value<0.05 was considered statistically significant.

## Subcutaneous xenograft tumor model and IVIS imaging

$5\times10^5$ luciferase-labeled HT29 cells were suspended in 0.1 ml serum-free DMEM and injected into the right back flank of 21- to 24-week-old NSG mice. Bioluminescence was measured once a week, by injection of 100 µl 15 mg/ml D-luciferin (LUCNA-1G, Goldbio) via the intra-orbital sinus. Mice were imaged using IVIS Lumina II (Advanced Molecular Vision). The acquired bioluminescence signals were normalized to the day 0 bioluminescence intensity.

## Statistics

All statistical analyses in our study were conducted using GraphPad Prism or R. Two-tailed t-test without equal variation assumption was used for data comparison between two experimental groups. For data comparison with at least three groups, one-way ANOVA test was performed first to assess the overall difference among groups. If differences existed, uncorrected Fisher's LSD test was performed to assess the significance of differences between two groups. Two-way repeated measures ANOVA and uncorrected Fisher's LSD test were used to assess the difference between datasets with time

series measurements, including growth curves of cell proliferation and in vivo BLI signals. p-Values less than 0.05 were considered significant.

## Acknowledgements

We thank Dr. Feng Cong for the generous gifts of reagents. We also thank Dr. Xuan Wang from the BCM Antibody-based Proteomics Core/Shared Resource for her technical assistance in performing reverse phase protein array; Ms. Xi-Lei Zeng and Ms. Xiaomin Yu from the Texas Medical Digestive Diseases Center (DDC) GEMS core for assistance in organoid culture; Ms. Cassandra Diegel from the Van Andel Institute for assistance. This work was financially supported by NIH/NCI (R01 CA204926 and CA271498 to YL), DOD-CDMRP (BC191649 and BC191646 to YL), and NIH-T32 grant (#1T32CA203690-01A1 to AK under Dr. Suzanne Fuqua). This project was also supported by the Breast Center Pathology Core as part of the SPORE program (P50 CA186784), Cytometry and Cell Sorting Core with funding from CPRIT (RP180672) and NIH (S10 RR024574), Antibody-based Proteomics Core with funding from CPRIT (RP210227) and NIH (S10 OD028648 to SH), DDC GEMS core with funding from NIH/NIDDK (DK56338), and resources from the Dan L Duncan Cancer Center with funding from NIH/NCI (P30 CA125123). The Van Andel Institute (VAI) provided additional support, including contributions from the VAI Vivarium (RRID:SCR_023211) and VAI Pathology and Biorepository Core (RRID:SCR_022912). PDS was supported by a post-doctoral fellowship from the American Cancer Society (PF-20-109-01), and MNM is supported by the NIH/NIDCR (K08DE3109).

## Additional information

### Funding

| Funder | Grant reference number | Author |
|---|---|---|
| National Institutes of Health | CA204926 | Yi Li |
| National Institutes of Health | CA271498 | Yi Li |
| Congressionally Directed Medical Research Programs | BC191649 | Yi Li |
| Congressionally Directed Medical Research Programs | BC191646 | Yi Li |
| National Institutes of Health | T32CA203690 | Amy T Ku |
| National Institutes of Health | CA186784 | Yi Li |
| National Institutes of Health | RR024574 | Yi Li |
| CPRIT | RP210227 | Shixia Huang |
| National Institutes of Health | OD028648 | Shixia Huang |
| National Institutes of Health | DK56338 | Yi Li |
| National Institutes of Health | CA125123 | Yi Li |
| American Cancer Society | PF-20-109-01 | Payton D Stevens |
| National Institutes of Health | K08DE3109 | Megan N Michalski |

| Funder | Grant reference number | Author |
|--------|------------------------|--------|

The funders had no role in study design, data collection and interpretation, or the decision to submit the work for publication.

## Author contributions

Fei Yue, Conceptualization, Data curation, Formal analysis, Validation, Methodology, Writing – original draft, Writing – review and editing; Amy T Ku, Payton D Stevens, Megan N Michalski, Weiyu Jiang, Jianghua Tu, Zhongcheng Shi, Yongchao Dou, Yi Wang, Galen Hostetter, Data curation; Xin-Hua Feng, Noah F Shroyer, Bing Zhang, Supervision; Xiangwei Wu, Shixia Huang, Resources; Bart O Williams, Qingyun Liu, Supervision, Writing – review and editing; Xia Lin, Conceptualization, Supervision, Writing – review and editing; Yi Li, Conceptualization, Resources, Supervision, Funding acquisition, Project administration, Writing – review and editing

## Author ORCIDs

Fei Yue ⓘ https://orcid.org/0000-0002-6057-0393
Amy T Ku ⓘ https://orcid.org/0000-0001-9398-7569
Jianghua Tu ⓘ https://orcid.org/0000-0002-6745-4897
Yi Li ⓘ https://orcid.org/0000-0002-9976-518X

Reviewer #1 (Public review): https://doi.org/10.7554/eLife.95639.3.sa1
Reviewer #2 (Public review): https://doi.org/10.7554/eLife.95639.3.sa2
Author response https://doi.org/10.7554/eLife.95639.3.sa3

# Additional files

## Supplementary files

MDAR checklist

## Data availability

All data are available in the main text or the supplementary materials.

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
