## [Editor Report · eLife Assessment]

This manuscript presents **solid** evidence suggesting that the loss of ZNRF3 and RNF43, two E3 ubiquitin ligases, leads to dysregulation of EGFR signaling in cancer. The authors propose that EGFR is a direct substrate of ZNRF3/RNF43. While the authors provide immunoprecipitation data showing increased detection of ubiquitinated species, this evidence does not definitively establish that EGFR itself is ubiquitinated by RNF43/ZNRF3. The absence of direct evidence for EGFR ubiquitination is a major limitation, although the findings are **useful** as they may provide novel insights into the mechanisms underlying EGFR-driven cancers and open new therapeutic avenues.

---

## [Referee Report · Reviewer #1 (Public review)]

Summary:

In this manuscript, the authors provide strong evidence that the cell surface E3 ubiquitin ligases RNF43 and ZNRF3, which are well known for their role in regulating cell surface levels of WNT receptors encoded by FZD genes, also target EGFR for degradation. This is newly identified function for these ubiquitin ligases beyond their role in regulating WNT signaling. Loss of RNF43/ZNRF3 expression leads to elevated EGFR levels and signaling, suggesting a potential new axis to drive tumorigenesis, whereas overexpression of RNF43 or ZNRF3 decreases EGFR levels and signaling. Furthermore, RNF43 and ZNRF3 directly interact with EGFR through their extracellular domains.

Strengths:

The data showing that RNF43 and ZNRF3 interact with EGFR and regulate its levels and activity are thorough and convincing, and the conclusions are largely supported.

Weaknesses:

Prior work established a clear role for RNF43 and ZNRF3 in regulating cell surface levels of FZD, a class of WNT receptors. These new findings that these E3 ubiquitin ligases also target EGFR add a new layer of complexity, and it remains unclear to what extent WNT signaling versus EGFR signaling are impacted in cancer settings. The authors acknowledge this gap in our understanding, which will likely be the topic of follow-up studies.

Comments on revisions:

The authors addressed my main concerns in this revised version and in their rebuttal comments. I have no further critiques to add.

---

## [Referee Report · Reviewer #2 (Public review)]

1st Public review:

Using proteogenomic analysis of human cancer datasets, Yu et al, found that EGFR protein levels negatively correlate with ZNFR3/RNF43 expression across multiple cancers. Interestingly, they found that CRC harbouring the frequent RNF43 G659Vfs*41 mutation exhibit higher levels of EGFR when compared to RNF43 wild-type tumors. This is highly interesting since this mutation is generally not thought to influence Frizzled levels and Wnt-bcatenin pathway activity. Using CRISPR knockouts and overexpression experiments, the authors show that EGFR levels are modulated by ZNRF3/RNF43. Supporting these findings modulation of ZNRF3/RNF43 activity using Rspondin also leads to increased EGFR levels. Mechanistically, the authors, show that ZNRF3/RNF43 ubiquitinate EGFR and lead to degradation. Finally, the authors present functional evidence that loss of ZNRF3/RNF43 unleashes EGFR-mediated cell growth in 2D culture and organoids and promote tumor growth in vivo.

Overall, the conclusions of the manuscript are well supported by the data presented, but some aspects of the mechanism presented need to be re-enforced to fully support the claims made by the authors. Additionally, the title of the paper suggests that ZNRF3 and RNF43 loss leads to hyperactivity of EGFR and that its signalling activity contribute to cancer initiation/progression. I don't think the authors convincingly showed this in their study.

Major points:

(1) EGFR ubiquitination. All of the experiments supporting that ZNFR3/RNF43 mediate EGFR ubiquitination are performed under overexpression conditions. A major caveat is also that none of the ubiquitination experiments are performed under denaturing conditions. Therefore, it is impossible to claim that the ubiquitin immunoreactivity observed on the western blots presented in Fig.4 corresponds to ubiquitinated-EGFR species.

Another issue is that in Figure 4A, the experiments suggest that the RNF43-dependent ubiquitination of EGFR is promoted by EGF. However, there is no control showing the ubiquitination of EGFR in the absence of EGF but under RNF43 overexpression. According to the other experiments presented in Figures 4B, 4C and 4F, there seems to be a constitutive ubiquitination of EGFR upon overexpression. How do the authors reconcile the role of ZNRF3/RNF43 vs c-cbl?

(2) EGFR degradation vs internalization. In Figure 3C, the authors show experiments that demonstrate that RNF43 KO increases steady state levels of EGFR and prevents its EGF-dependent proteolysis. Using flow cytometry they then present evidence that the reduction in cell surface levels of EGFR mediated by EGF is inhibited in the absence of RNF43. The authors conclude that this is due to inhibition of EGF-induced internalization of surface EGF. However, the experiments are not designed to study internalization and rather merely examine steady state levels of surface EGFR pre and post treatment. These changes are an integration of many things (retrograde and anterograde transport mechanisms presumable modulated by EGF). What process(es) is/are specifically affected by ZNFR3/RNF43? Are these processes differently regulated by c-cbl? If the authors are specifically interested in internalization/recycling, the use of cell surface biotinylation experiments and time courses are needed to examine the effect of EGF in the presence or absence of the E3 ligases.

(3) RNF43 G659fs*41. The authors make a point in Figure 1D that this mutant leads to elevated EGFR in cancers but do not present evidence that this mutant is ineffective in mediated ubiquitination and degradation of EGFR. As this mutant maintains its ability to promote Frizzled ubiquitination and degradation, it would be important to show side by side that it does not affect EGFR. This would perhaps imply differential mechanisms for these two substrates.

(4) "Unleashing EGFR activity". The title of the paper implies that ZNRF3/RNF43 loss leads to increased EGFR expression and hence increased activity that underlies cancer. However, I could find only one direct evidence showing that increased proliferation of the HT29 cell line mutant for RNF43 could be inhibited by the EGFR inhibitor Erlotinib. All the other evidence presented that I could find is correlative or indirect (e.g. RPPA showing increased phosphorylation of pathway members upon RNF43 KO, increased proliferation of a cell line upon ZNRF3/ RNF43 KO, decreased proliferation of a cell line upon ZNRF3/RNF43 OE in vitro or in xeno...). Importantly, the authors claim that cancer initiation/ progression in ZNRF3/RNF43 mutant may in some contexts be independent of their regulation of Wnt-bcatenin signaling and relying on EGFR activity upregulation. However, this has not been tested directly. Could the authors leverage their znrf3/RNF43 prostate cancer model to test whether EGFR inhibition could lead to reduced cancer burden whereas a Frizzled or Wnt inhibitor does not?

More broadly, if EGFR signaling were to be unleashed in cancer, then one prediction would be that these cells would be more sensitive to EGFR pathway inhibition. Could the authors provide evidence that this is the case? Perhaps using isogenic cell lines or a panel of patient derived organoids (with known genotypes).

Comments on revisions:

The most important criticism of this manuscript that I raised in my original review has not been addressed. Indeed, the authors claim that EGFR is a direct substrate of the RNF43/ZNFR3 E3 ligase. This has not been directly demonstrated. Indeed, showing increased detection of ubiquitinated species in an immunoprecipitate could mean that a protein is directly modified. However, an alternative explanation is that a protein that is co-immunoprecipitated with the target protein is ubiquitinated (such as several EGFR adapters and interacting partners). Performing these experiments under denaturing conditions is one way to determine that EGFR is the substrate. Alternatively, a quantitative MS approach to quantify an increase in ubiquitinated peptides would also enable the authors to conclude that EGFR is indeed a substrate.

In addition, one of the main conclusions of the authors is that EGFR activity is unleashed in cancer following ZNRF3 and/or RNF43 loss (as the title suggests). There is still no direct evidence in the manuscript that this is the case. I appreciate the new data showing that MEF with knockout of RNF43/ZNRF3 are sensitive to EGFR inhibitor (and not porcupine inhibitor) but what is the data supporting that EGFR activity is "unleashed" in cancer? The authors still claim that ZNRF3 and RNF43 loss could impact cancer initiation/development in a Wnt-independent fashion (see lines 341-343). I believe this conclusion is based on correlative staining of nuclear bcatenin (which is in itself not a reliable readout of active sginaling) and not on functional data.... I suggested in my original review that the authors should test the efficacy of EGFR inhibitor and Wnt inhibitor in the prostate cancer model that they present in Figure 7 that would have enabled them to firmly conclude about their relative contribution. This was largely handwaved in their rebuttal letter... Doing experiment in WT cells is not the same as addressing this question in the context of cancer.

Finally, the authors use CRISPR KO experiments, without assessing editing or KO efficiencies throughout the manuscript and simply assume that the gRNA work. In my opinion this is an unacceptable practice.

---

## [Author Response]

The following is the authors’ response to the original reviews.

**Public Reviews:**
**Reviewer #1 (Public Review)**:Summary:In this manuscript, the authors provide strong evidence that the cell surface E3 ubiquitin ligases RNF43 and ZNRF3, which are well known for their role in regulating cell surface levels of WNT receptors encoded by FZD genes, also target EGFR for degradation. This is a newly identified function for these ubiquitin ligases beyond their role in regulating WNT signaling. Loss of RNF43/ZNRF3 expression leads to elevated EGFR levels and signaling, suggesting a potential new axis to drive tumorigenesis, whereas overexpression of RNF43 or ZNRF3 decreases EGFR levels and signaling. Furthermore, RNF43 and ZNRF3 directly interact with EGFR through their extracellular domains.Strengths:The data showing that RNF43 and ZNRF3 interact with EGFR and regulate its levels and activity are thorough and convincing, and the conclusions are largely supported.Weaknesses:While the data support that EGFR is a target for RNF43/ZNRF3, some of the authors' interpretations of the data on EGFR's role relative to WNT's roles downstream of RNF43/ZNRF3 are overstated. The authors, perhaps not intentionally, promote the effect of RNF43/ZNRF3 on EGFR while minimizing their role in WNT signaling. This is the case in most of the biological assays (cell and organoid growth and mouse tumor models). For example, the conclusion of "no substantial activation of Wnt signaling" (page 14) in the prostate cancer model is currently not supported by the data and requires further examination. In fact, examination of the data presented here indicates effects on WNT/b-catenin signaling, consistent with previous studies.Cancers in which RNF43 or ZNRF3 are deleted are often considered to be "WNT addicted", and inhibition of WNT signaling generally potently inhibits tumor growth. In particular, treatment of WNT-addicted tumors with Porcupine inhibitors leads to tumor regression. The authors should test to what extent PORCN inhibition affects tumor (and APC-min intestinal organoid) growth. If the biological effects of RNF43/ZNRF3 loss are mediated primarily or predominantly through EGFR, then PORCN inhibition should not affect tumor or organoid growth.

We thank the reviewer’s appreciation of the key strength of our study. We fully agree with the reviewer that RNF43/ZNRF3 play key roles in restraining WNT signaling and their deletions activate WNT signaling that leads to cancer promotion, as discussed and cited in our manuscript (Hao et al, 2012; Koo et al, 2012). We have revised the language in this manuscript to avoid any confusion or appearance of downplaying this known signaling pathway in cancer progression.

What we would like to highlight in this work is that our study uncovered an effect of RNF43/ZNRF3 on EGFR, leading to biological impact in multiple model systems. In particular, we included the APC-mutated human cancer cell line HT29 and Apc min mouse intestinal tumor organoids. In the context of APC mutations, β-catenin stabilization and the activation of WNT target genes are essentially decoupled from upstream WNT ligand binding to WNT receptors, thus we could primarily focus on the effect of RNF43/ZNRF3 on EGFR. Our statement of “no substantial activation of WNT signaling” as cited by the reviewer was made in describing the data in Fig. 7E where we did not observe β-catenin accumulation in the nucleus and reasoned no substantial activation of canonical WNT signaling. We agree that further examination would help strengthen the conclusion and appreciate the reviewer’s suggestion of PORCN inhibition experiments. While PORCN inhibition is a valuable experiment in models with abundance of WNT ligands/receptors and non-mutationally activated regulators of WNT signaling (Yu et al, 2020), in biological scenarios with existing APC mutations, another group has previously demonstrated that PORCN inhibition had no observable effect on WNT signaling in APC-deficient cells (PMID: 29533772). In our initial submission, we confirmed this predicted low response to manipulation of WNT signaling components upstream of a mutated APC. We showed that addition of RSPO1 *in Apc min* mouse intestinal tumor organoids failed to further activate WNT target expression (Fig. 6G). Furthermore, in this revised manuscript, we added new data on EGFR inhibition and PORCN inhibition in WT and *Znrf3* KO MEFs (Fig. 6L). PORCN inhibition had no impact on cell growth in neither WT nor *Znrf3* KO MEFs, suggesting that *Znrf3* KO promoting MEF growth is WNT independent. In contrast, inhibition of EGFR downstream signaling components (Fig. 6L) significantly blocked MEF growth and abolished the impact of *Znrf3* KO in MEF growth. This new evidence further supports our main conclusion that RNF43/ZNRF3 controls EGFR signaling to regulate cell growth.

**Reviewer #2 (Public Review):**
Using proteogenomic analysis of human cancer datasets, Yu et al, found that EGFR protein levels negatively correlate with ZNFR3/RNF43 expression across multiple cancers. Interestingly, they found that CRC harbouring the frequent RNF43 G659Vfs*41 mutation exhibits higher levels of EGFR when compared to RNF43 wild-type tumors. This is highly interesting since this mutation is generally not thought to influence Frizzled levels and Wnt-bcatenin pathway activity. Using CRISPR knockouts and overexpression experiments, the authors show that EGFR levels are modulated by ZNRF3/RNF43. Supporting these findings, modulation of ZNRF3/RNF43 activity using Rspondin also leads to increased EGFR levels. Mechanistically, the authors, show that ZNRF3/RNF43 ubiquitinate EGFR and leads to degradation. Finally, the authors present functional evidence that loss of ZNRF3/RNF43 unleashes EGFR-mediated cell growth in 2D culture and organoids and promotes tumor growth in vivo.

Overall, the conclusions of the manuscript are well supported by the data presented, but some aspects of the mechanism presented need to be reinforced to fully support the claims made by the authors. Additionally, the title of the paper suggests that ZNRF3 and RNF43 loss leads to the hyperactivity of EGFR and that its signalling activity contributes to cancer initiation/progression. I don't think the authors convincingly showed this in their study.

We thank the reviewer commenting that our “conclusions of the manuscript are well supported by the data presented.” We address the concerns raised by this reviewer in an itemized way as detailed below:

Major points:(1) EGFR ubiquitination. All of the experiments supporting that ZNFR3/RNF43 mediates EGFR ubiquitination are performed under overexpression conditions. A major caveat is also that none of the ubiquitination experiments are performed under denaturing conditions. Therefore, it is impossible to claim that the ubiquitin immunoreactivity observed on the western blots presented in Figure 4 corresponds to ubiquitinated-EGFR species. Another issue is that in Figure 4A, the experiments suggest that the RNF43-dependent ubiquitination of EGFR is promoted by EGF. However, there is no control showing the ubiquitination of EGFR in the absence of EGF but under RNF43 overexpression. According to the other experiments presented in Figures 4B, 4C, and 4F, there seems to be a constitutive ubiquitination of EGFR upon overexpression. How do the authors reconcile the role of ZNRF3/RNF43 vs c-cbl?

We agree with this reviewer of the limitation of overexpression experiments. In this manuscript, we actually leveraged both overexpression and knockout systems to demonstrate that ZNRF3/RNF43 regulates EGFR ubiquitination: in Fig 4A, we showed that overexpression of RNF43 increased EGFR ubiquitination; in Fig 4B&C and Fig S3A, we showed that RNF43 knockout decreased EGFR ubiquitination; in Fig 4F, we showed that overexpression of ZNRF3 WT increased EGFR ubiquitination but overexpression of ZNRF3 RING domain deletion mutant failed to increase EGFR ubiquitination.

We also appreciate the rigor with which the reviewer has approached our methodology. We acknowledge that denaturing conditions can provide additional validation, but the technical challenges associated with denaturing conditions include the potential disruption of epitope structures recognized by these antibodies. Our methodology was chosen to balance the need for accurate detection with the preservation of protein structure and function, which are crucial for understanding the biological implications of EGFR ubiquitination. Moreover, our immunoprecipitation and subsequent Western blotting were stringent with high SDS and 2-ME, optimized to minimize non-specific binding and enhance the specificity of detection. We believe that the data presented are robust and contribute significantly to the existing body of knowledge on EGFR ubiquitination.

CBL is a well-known E3 ligase of EGFR, and it induces EGFR ubiquitination upon EGF ligand stimulation. Therefore, in order to have a fair comparison of RNF43 and CBL on EGFR ubiquitination, we designed Fig 4A and related experiments in the setting of EGF stimulation. We observed that RNF43 overexpression increased EGFR ubiquitination as potently as CBL did. Following this result, we further demonstrated that knockout of *RNF43* decreased endogenous ubiquitinated EGFR level in the unstimulated/basal condition (Fig 4B) as well as in the EGF-stimulated condition (Fig 4C). We acknowledge the importance and interest in fully understanding how ZNRF3/RNF43 interplays with the functions of CBL in regulating EGFR ubiquitination. This line of investigation indeed holds the potential to uncover novel regulatory mechanisms in detail. However, the primary focus of the current study was to establish a foundational understanding of ZNRF3/RNF43 role in regulating EGFR ubiquitination. We look forward to exploring further in future work.

(2) EGFR degradation vs internalization. In Figure 3C, the authors show experiments that demonstrate that RNF43 KO increases steady-state levels of EGFR and prevents its EGF-dependent proteolysis. Using flow cytometry they then present evidence that the reduction in cell surface levels of EGFR mediated by EGF is inhibited in the absence of RNF43. The authors conclude that this is due to inhibition of EGF-induced internalization of surface EGF. However, the experiments are not designed to study internalization and rather merely examine steady-state levels of surface EGFR pre and post-treatment. These changes are an integration of many things (retrograde and anterograde transport mechanisms presumable modulated by EGF). What process(es) is/are specifically affected by ZNFR3/RNF43? Are these processes differently regulated by c-cbl? If the authors are specifically interested in internalization/recycling, the use of cell surface biotinylation experiments and time courses are needed to examine the effect of EGF in the presence or absence of the E3 ligases.

We agree that our study design primarily assesses EGFR levels on the cell surface before and after EGF treatment and does not comprehensively measure the whole internalization process. In response to the reviewer’s comments, we have revised the relevant sections of manuscript to clarify that our current findings are focused on changes in cell surface EGFR and do not extend to the detailed mechanisms of EGF-induced internalization or recycling.

(3) RNF43 G659fs*41. The authors make a point in Figure 1D that this mutant leads to elevated EGFR in cancers but do not present evidence that this mutant is ineffective in mediated ubiquitination and degradation of EGFR. As this mutant maintains its ability to promote Frizzled ubiquitination and degradation, it would be important to show side by side that it does not affect EGFR. This would perhaps imply differential mechanisms for these two substrates.

Fig 1D is based on bioinformatic analysis of colon cancer patient samples, showing that RNF43 G659Vfs*41 mutant tumors exhibited significantly higher levels of EGFR protein compared to RNF43 WT tumors. Following this lead, we investigated whether this *RNF43 G659fs*41* hotspot mutation lost its role in downregulating EGFR. To this end, we transfected the same amount of control vector, RNF43 WT, RING deletion mutant, G659fs*41 mutant DNA into 293T cells and measured the level of EGFR (co-transfected). As shown in Author response image 1, overexpression of RNF43 WT decreased EGFR level while overexpression of RING deletion mutant had no impact on EGFR level as compared with the Vector group, which is consistent with our findings in the manuscript. Cells transfected with the RNF43 G659Vfs*41 mutant exhibited nearly normal levels of EGFR; however, we also observed that RNF43 G659Vfs*41 was less expressed than WT, even though the same amounts of DNA were transfected. Therefore, the insubstantial impact on EGFR levels could be attributed to both functional loss or compromised stability of *RNF43 G659Vfs*41* mRNA or protein. Further investigation on *RNF43 G659Vfs*41* mRNA and protein stability vs. RNF43 G659Vfs*41 protein function is needed to draw a solid conclusion.

(4) "Unleashing EGFR activity". The title of the paper implies that ZNRF3/RNF43 loss leads to increased EGFR expression and hence increased activity that underlies cancer. However, I could find only one direct evidence showing that increased proliferation of the HT29 cell line mutant for RNF43 could be inhibited by the EGFR inhibitor Erlotinib. All the other evidence presented that I could find is correlative or indirect (e.g. RPPA showing increased phosphorylation of pathway members upon RNF43 KO, increased proliferation of a cell line upon ZNRF3/ RNF43 KO, decreased proliferation of a cell line upon ZNRF3/RNF43 OE in vitro or in xeno...). Importantly, the authors claim that cancer initiation/ progression in ZNRF3/RNF43 mutants may in some contexts be independent of their regulation of Wnt-bcatenin signaling and relying on EGFR activity upregulation. However, this has not been tested directly. Could the authors leverage their znrf3/RNF43 prostate cancer model to test whether EGFR inhibition could lead to reduced cancer burden whereas a Frizzled or Wnt inhibitor does not?More broadly, if EGFR signaling were to be unleashed in cancer, then one prediction would be that these cells would be more sensitive to EGFR pathway inhibition. Could the authors provide evidence that this is the case? Perhaps using isogenic cell lines or a panel of patient-derived organoids (with known genotypes).

We appreciate the reviewer’s suggestion to provide more direct evidence demonstrating the importance of the ZNRF3/RNF43-EGFR axis in cancer cell proliferation. In this revised manuscript, we further studied this issue in the WT vs. *Znrf3* KO MEF cells. We observed that treatment with the EGFR inhibitor erlotinib did not affect WT MEF but stunted the growth advantage of *Znrf3* KO MEF cells (Fig. 6L). On the other hand, treatment with the porcupine inhibitor C59 did not impact either WT or *Znrf3* KO MEF cells (Fig. 6L), suggesting a more important role of the ZNRF3/RNF43-EGFR axis in mediating the enhanced cell growth of MEF caused by *Znrf3* knockout. Furthermore, considering EGFR is often mutated in human cancer, to increase the clinical relance of our study, we also tested the effect of RNF43 knockout on EGFR L858R (Fig. 2D), a common oncogenic EGFR mutant, and found that *RNF43* knockout in HT29 boosted levels of this EGFR mutant detected by its FLAG tag, suggesting that RNF43 degrades both WT and mutated EGFR and its loss can enhance signaling of both WT EGFR and its oncogenic mutant . However, we emphasize again that this manuscript is in no way written to diminish the proven importance of ZNRF3/RNF43-WNT-β-catenin axis in cancer and development.

**Recommendations for the authors:**

**Reviewer #1 (Recommendations For The Authors):**
The main conclusion that EGFR is targeted for degradation by RNF43 and ZNRF3 is well supported and documented. Figures 1-5 and associated supplemental figures contain largely convincing data. Figures 6 and 7, however, require some modifications, as follows in order of appearance:Figure 6C: Growth of intestinal tumor organoids from Apcmin mice does not require Rspo, however, the authors show that these organoids grow larger in the presence of Rspo, an effect they attribute to increased EGFR activity, rather than increased WNT activity. While this conclusion may be correct, the authors should address this possibility by treating the organoids with PORCN inhibitor. The prediction would be that Rspo treatment still increases organoid size in the presence of PORCN inhibition. A further prediction would be that blocking EGFR (e.g. with Cetuximab) will abrogate the RSPO1 effect.

Yes, we attributed the impact of Rspo on Apc min organoid growth to enhanced EGFR activity because we observed increased EGFR levels (Fig 6F) but no detectable increase in eight WNT target genes assayed. We agree that further pharmacologic experiments would further boost our conclusion, but our few attempts at treating organoids encountered technical difficulties. Hence, we switched to testing PORCN inhibition vs EGFR inhibition in WT and *Znfr33* KO MEFs. As shown in the revised Fig. 6L, EGFR inhibition significantly reversed the growth advantage caused by *Znrf3* KO but C59 did not.

Figure 6G: It is unclear why the authors provide "8-day RSPO1 treatment" data. Here, EGFR mRNA appears to be elevated 2-fold (perhaps not statistically significant), and the Wnt targets Lef1 and Axin2 are decreased, as indicated by the statistical significance. What point is being made here?

Our observation of increased size of APC min mouse intestinal tumor organoids and increased the EGFR protein levels were at 8 days of RSPO1 treatment. Therefore, we measured mRNA levels at the same time point with the 2-day time point also included for comparison. The goal of this qPCR experiment was to detect the contribution of WNT signaling, and we did not detect an increased transcriptional readout. We included EGFR mRNA levels for comparison, and we did not detect a statistically significant increase, consistent with our experiments concluding that ZNRF3/RNF43 regulate EGFR at the protein level. As stated in the preceding response, these data led us to attribute the impact of Rspo on Apc min organoid growth to enhanced EGFR activity.

Figure 7A: This requires quantitation. How many mice were used per cell line? The data shown is not particularly convincing, with ZNRF3 overexpressing HT29 cells growing detectably. Showing representative mice is fine, but this should be supplemented with quantitation of all mice.

We had provided this data. The BLI signal quantification was shown below the representative BLI images. Seven mice were used per cell line, as annotated at the top of the graph.

Figure 7B: The authors assert that "canonical WNT signaling, based on levels of active-β-Catenin (non-phosphorylated at Ser33/37/Thr41; Figure 7B), remained unaffected". As shown, 2 of the 3 Myc-Znrf3 tumors have increased active-b-catenin signal over the GFP tumors. This indicates to me that canonical Wnt signaling was affected. The authors either need to present quantitative data that supports this claim or modify their conclusions. As presented, I don't think it is appropriate to decouple the effect of Znrf3 overexpression on EGFR from its effect on WNT.

As requested, we have quantified the level of non-phospho β-Catenin at Ser33/37/Thr41 and found no significant differences (p > 0.05) between the control group vs. ZNRF3 overexpression group. We once again note that our manuscript was not meant to dispute the proven signaling and biological significance of WNT signaling regulation by ZNRF3/RNF43, and we have proof-read the manuscript multiple times to ensure that we did not make any generalized or misleading statements in this aspect.

**Author response image 2. sa3fig2:** 

Figure 7E: Here the authors assert that "no substantial activation of canonical Wnt signaling" in the Z&R KO tumors, however, the figure shows a substantial increase in active b-catenin staining. The current resolution is insufficient to claim that there is no increase in nuclear b-catenin. The authors' claim that WNT signaling is not involved here is not supported by the data presented here. One way to demonstrate that this effect is through EGFR activation and not through WNT activation is to treat mice with PORCN inhibitor. WNT-addicted tumors, such as by Rnf43 or Znrf3 deletion, regress upon PORCN inhibition. In this case, if the effect of Z&R KO is mediated through EGFR rather than WNT, then there should be no effect on tumor growth upon PORCN inhibition. This is a critical experiment in order to make this point.

We appreciate the reviewer’s comments and suggestion of experiments. We based our initial statement on insubstantial nuclear β-catenin staining, but we agree that immunohistochemical staining lacks the resolution suitable for quantification. We could not generate the adequate number of KO animals for these in vivo experiments in the window of time planned for this revision. Rather, as shown in the newly added Fig. 6L, we tested EGFR inhibition and PORCN inhibition in *Znrf3* KO MEFs and obtained strong data further supporting EGFR in mediating *Znrf3* KO promotion of MEF growth. Notwithstanding, we have carefully revised our description of the in vivo data in Fig 7E to avoid any confusion or over-interpretation.

Minor points:Figure 2A: provide quantitation of this immunoblot.

We have revised manuscript with quantification result shown next to the immunoblot.

Figure 2B: provide more detail in the figure legend and in the Materials and Methods section on how the KO MEFs were generated. Confirmation that Znrf3 (or in cases of Rnf43 KO) expression is lost in KO would be advisable.

We have confirmed *Znrf3* KO by genotyping and *RNF43* KO by immunofluorescent staining. We have also tested multiple commercial anti-ZNRF3 antibodies and anti-RNF43 antibodies for Western blotting, but they all failed.

Figure 4C is a little misleading. The schematic indicates that ECD-TM and TM-ICD truncations were analyzed for both ZNRF3 and RNF43. However, Figure 4 only shows data for ZNRF3, and the corresponding Figure S4 lacks data for the TM-ICD of Rnf43. A recommendation is to show only those schematics for which data is presented in that figure. On a related topic, the results using the deltaRING constructs (Figure S5) are not mentioned/described in the text.

We think that the reviewer meant Fig 5C. We have revised the Fig 5C by removing the RNF43 label, and we confirm that Results section does include the data in Fig S5.

Figure S4A: Only ZNRF3 is indicated in this figure. Please explain why RNF43 is not represented here. Also, indicate what is plotted along the x-axis.

We only detected the endogenous ZNRF3-EGFR interaction, possibly because the RNF43 protein level is relatively low in the cell line we used for the mass spec experiment. X-axis is the proteins ordered based on Y-axis values as detailed in the figure legend -- each data point was arranged along the x axis based on the fold change of iBAQ of EGFR-associated proteins identified in EGF-stimulated vs. control in the log2 scale, from low to high (from left to right on x axis). We have added the phrase “Proteins detected by Mass-Spec” for X-axis.

**Reviewer #2 (Recommendations For The Authors):**
Minor Points.(1) In Figure 2B, the authors claim that Znrf3 KO enhanced both EGFR and p-EGFR levels both in the absence and presence of EGF. Although it is clear in the presence of EGF, the increased in p-EGFR in the absence of EGF is less than clear.

We have revised the manuscript to more clearly state the result in Fig 2B.

(2) Importantly the authors validated their findings using three independent RNF43 gRNA (fig S2D) but they do not show the editing efficiency obtained with the gRNA.

We did not include RNF43 IB in this Figure due to lack of specific antibodies for detecting RNR43 in IB. We have no reasons to doubt adequate efficiency of knockout since EGFR was increased compared to the control group. As a result, we did not perform deep sequencing to validate knockout efficacy.

(3) In S2E, the authors show that KO of either ZNRF3 or RNF43 enhance HER2 levels. This suggests that there is no redundancy between these E3 ligases, at least in this context. How do the authors reconcile that?

The reviewer raised an interesting issue. Due to the lack of WB antibodies for these two proteins, we would not easily assess the feedback impact of knockout of either gene on the protein levels of the other gene. We speculate that there may be a threshold level of the sum of the two proteins that is needed for adequate degradation of HER2, leading to HER2 increase when either gene is knocked out. Detailed studies of this issue is beyond the scope of this current work.

(4) Experiments performed in Fig 3C are performed in only one clone. The authors need to repeat in an additional clone or rescue this phenotype using a RNF43 cDNA.

Our RNF43 KO HT29 line is a pool of KO cells, not a single clone.

(5) In Figure 7E, the authors suggest that the absence of nuclear bcatenin means that canonical Wnt signaling is unaffected. It is widely known that nuclear bcatenin is often not correlating with pathway activity.

As stated above, we have revised the manuscript to avoid confusion and misinterpretation.

(6) What is the nature of the error bars in Fig 3c? Are the differences statistically significant?

As mentioned in the figure legend, the error bars are SEM. The result is statistically significant, and p-value is noted in the graph.

(7) In the Figure legends, it should be stated clearly how many biological replicates were performed for each experiment and single data points should be plotted where applicable (e.g. qPCR data). It would be helpful if the uncropped and unprocessed Western blot membranes and replicates that are not shown would be accessible to allow the reader a more comprehensive view of the acquired data, especially for blots that were quantified (e.g. Figure 2F, Figure 3C, there is clearly some defect on the blot).For WB representation, it would be helpful to include more size markers on the Western blots (especially on the Ips that show ubiquitin smear) and in general to use a reference protein (GAPDH, Actin, Vinculin) that is closer to the protein being accessed.More details should be added in the Methods section to explain how protocols were performed in detail. For example, it should be explained how the viruses used for infecting cells were produced (which plasmids were transfected using which transfection reagent, how long was the virus collected for, etc). Then, it should be stated how long the cells were undergoing selection before being harvested. Because the expression of the viral constructs potentially has an effect on cell proliferation through EGFR, this information is quite relevant. This is just an example, there are details missing in nearly every section (Flow: washing protocols, gating protocols (Live/dead stain?), WB: RIPA lysis buffer composition? How much protein was loaded on blots? How was protein quantification done? IP: how were washes performed and how often repeated?)Missing: antibody dilutions for IF, IHC, and WB, plasmid backbones, sequences and availability, qPCR primer sequences from Origene.Incucyte experiments are not described.

We have revised the relevant sections to include more details.

(8) Line 141: revise text: 2x mRNA abundance in the same sentence.Line 162: define intermediate expression better.Line 197/198: revise text ('the predominant one'?).Line 218/219: revise text (Internalisation of surface EGFR?).Line 245: clarify in text that it is endogenous EGFR that is being pulled down.Line 264: typo: conserved instead of conservative.Line 324: revise text (What does 'unknown significance' mean).Line 396/397: revise text: 2x Co-IP in the same sentence.Figure 3 D/E: more details on the Method in the figure legend.

We have revised them accordingly.